# Single-molecule FRET unveils induced-fit mechanism for substrate selectivity in flap endonuclease 1

Fahad Rashid[1], Paul D Harris[1†], Manal S Zaher[1†], Mohamed A Sobhy[1], Luay I Joudeh[1], Chunli Yan[2,3], Hubert Piwonski[1], Susan E Tsutakawa[4], Ivaylo Ivanov[2,3], John A Tainer[4,5], Satoshi Habuchi[1], Samir M Hamdan[1*]

[1]Division of Biological and Environmental Science and Engineering, King Abdullah University of Science and Technology, Thuwal, Saudi Arabia; [2]Department of Chemistry, Georgia State University, Atlanta, United States; [3]Center for Diagnostics and Therapeutics, Georgia State University, Atlanta, United States; [4]Lawrence Berkeley National Laboratory, Berkeley, United States; [5]Department of Molecular and Cellular Oncology, University of Texas MD Anderson Cancer Center, Houston, United States

*For correspondence: samir.
hamdan@kaust.edu.sa

†These authors contributed
equally to this work

Competing interests: The
authors declare that no
competing interests exist.

Reviewing editor: James M
Berger, Johns Hopkins University
School of Medicine, United
States

**Abstract** Human flap endonuclease 1 (FEN1) and related structure-specific 5'nucleases precisely identify and incise aberrant DNA structures during replication, repair and recombination to avoid genomic instability. Yet, it is unclear how the 5'nuclease mechanisms of DNA distortion and protein ordering robustly mediate efficient and accurate substrate recognition and catalytic selectivity. Here, single-molecule sub-millisecond and millisecond analyses of FEN1 reveal a protein-DNA induced-fit mechanism that efficiently verifies substrate and suppresses off-target cleavage. FEN1 sculpts DNA with diffusion-limited kinetics to test DNA substrate. This DNA distortion mutually 'locks' protein and DNA conformation and enables substrate verification with extreme precision. Strikingly, FEN1 never misses cleavage of its cognate substrate while blocking probable formation of catalytically competent interactions with noncognate substrates and fostering their pre-incision dissociation. These findings establish FEN1 has practically perfect precision and that separate control of induced-fit substrate recognition sets up the catalytic selectivity of the nuclease active site for genome stability.

## Introduction

Biologically-critical, structure-specific 5'nucleases are highly conserved endo- or exo-nucleases that hydrolyze phosphodiester bonds that are one nucleotide into the 5'end of single-stranded(ss)/double-stranded(ds)-DNA junctions (*Figure 1A*), including nicks, gaps, flaps, bubbles and four-way junctions (*Balakrishnan and Bambara, 2013*; *Finger et al., 2012*; *Tsutakawa et al., 2014*; *Tsutakawa and Tainer, 2012*). This conserved cleavage site despite diverse structures operates by uniformly binding to a bent junction to place the scissile phosphate near the active site (*Liu et al., 2015*; *Orans et al., 2011*; *Tsutakawa et al., 2011*) (*Figure 1B*). Yet, the mechanism underlying this specificity remains unclear including the question of whether 5'nucleases actively distort the DNA or selectively bind to a DNA that bends spontaneously (*Craggs et al., 2014*; *Sobhy et al., 2013*). Such mechanistic knowledge not only pertains to biological understanding but also to strategies for the design of specific inhibitors as potential cancer drugs (*Exell et al., 2016*). Catalysis is proposed to require changes in the protein conformation that assembles the active site (*Devos et al., 2007*; *Lee et al., 2015*; *Liu et al., 2015*; *Orans et al., 2011*; *Sakurai et al., 2005*; *Tsutakawa et al., 2011*)

**eLife digest** When a cell divides it must copy its genetic information, which is found in the form of strands of DNA. Damage to the DNA may lead to cancer or a number of genetic diseases. However, every time a cell divides more than 10 million toxic "flaps" of excess DNA are generated. A protein called flap endonuclease 1 (FEN1) keeps the DNA in good repair by cutting off the flaps in a highly specific and selective manner.

Many proteins that interact with DNA are attracted to specific genetic sequences within the DNA strands. However, this is not the case for FEN1 and several other "structure-specific" proteins that help to repair and replicate DNA strands. So how do these proteins select the correct regions of DNA to interact with?

Rashid et al. used single-molecule fluorescence measurements to examine how purified FEN1 proteins interact with DNA flaps. The results show that FEN1 can perfectly recognize and correctly remove flaps through a process called "mutual-induced fit", where the DNA and FEN1 are shaped by each other to produce a highly specific structure.

Further work is now needed to examine whether other proteins that are related to FEN1 use a similar process to ensure that they always cut DNA in the same way. More detailed and direct examination of the structure of FEN1 through other experimental methods may also help to reveal how the mutual-induced fit process enables FEN1 to achieve such high levels of precision. This could increase our understanding of how problems with FEN1 and similar proteins lead to different genetic diseases.

and movement of the scissile phosphate closer to the catalytic metals (*Liu et al., 2015*; *Orans et al., 2011*; *Tsutakawa et al., 2011*). Although possible steps in the substrate selection and cleaving process have been described (*Devos et al., 2007*; *Lee et al., 2015*; *Liu et al., 2015*; *Orans et al., 2011*; *Sakurai et al., 2005*; *Tsutakawa et al., 2011*), much of the control mechanisms by DNA and protein conformational changes that lead to exquisite catalytic selectivity and efficiency remain controversial and largely undetermined.

Flap endonuclease 1 (FEN1) and its substrate and product complexes provide a prototypic system for unveiling the extreme catalytic selectivity of structure-specific 5'nucleases. Whereas sequence-based specificity partially explains the secret of replication fidelity, key information is missing about the basis for the structure-based excision required at more than 10 million Okazaki fragment sites during human DNA replication. Strikingly, FEN1 maintains exquisite specificity with extreme efficiency that enhances the hydrolysis rate of target phosphodiester bonds by $\sim 10^{17}$, and in vitro reaction rates resemble those of enzyme-substrate encounters (*Finger et al., 2009*). FEN1 recognizes dsDNA bearing a double-flap (DF) nick junction consisting of short ssDNA or ssRNA 5'flaps and strictly one nucleotide (nt) ssDNA 3'flap (*Figure 1A*) (*Finger et al., 2009*; *Kao et al., 2002*). DF intermediates are produced during Okazaki fragment synthesis on the lagging strand and during long-patch base excision repair (*Garg et al., 2004*; *Liu et al., 2005*). Mutations that reduce FEN1 expression or alter its activity are linked to cancers and genetic diseases (*Balakrishnan and Bambara, 2013*; *Henneke et al., 2003*; *Kucherlapati et al., 2002*; *Zheng et al., 2007*, *2011*). In cells, the 5'flap is complementary to the template strand, enabling the junction to equilibrate and form a single nt 3'flap (*Figure 1A*). Upon 5'flap cleavage, the 3'flap complements the newly unpaired template base to create a DNA ligase 1 sealable nick (*Figure 1A*). FEN1 contacts duplex DNA from both sides of the flap junction through a 100° bend stabilized by the interaction of the superfamily-conserved hydrophobic wedge with the junction (*Figure 1B*) (*Tsutakawa et al., 2011*). A superfamily-conserved helical gateway covered by a unique FEN1 helical cap forms a narrow cavity at the DNA junction. This gateway is suitable to select for threading ss 5'flaps with a free end (*Figure 1B*) (*Gloor et al., 2010*; *Patel et al., 2012*; *Sobhy et al., 2013*; *Tsutakawa et al., 2011*). Alternatively, clamping the 5'flap away from the active site is a proposed selection mechanism (*Orans et al., 2011*). A small cavity makes contacts with the 3'flap and may impose specificity for the single nt 3'flap (*Figure 1B*). Part of the cap-helical gateway, which contains catalytically indispensable residues, and the 3'flap-binding pocket appear disordered without DNA (*Sakurai et al., 2005*),

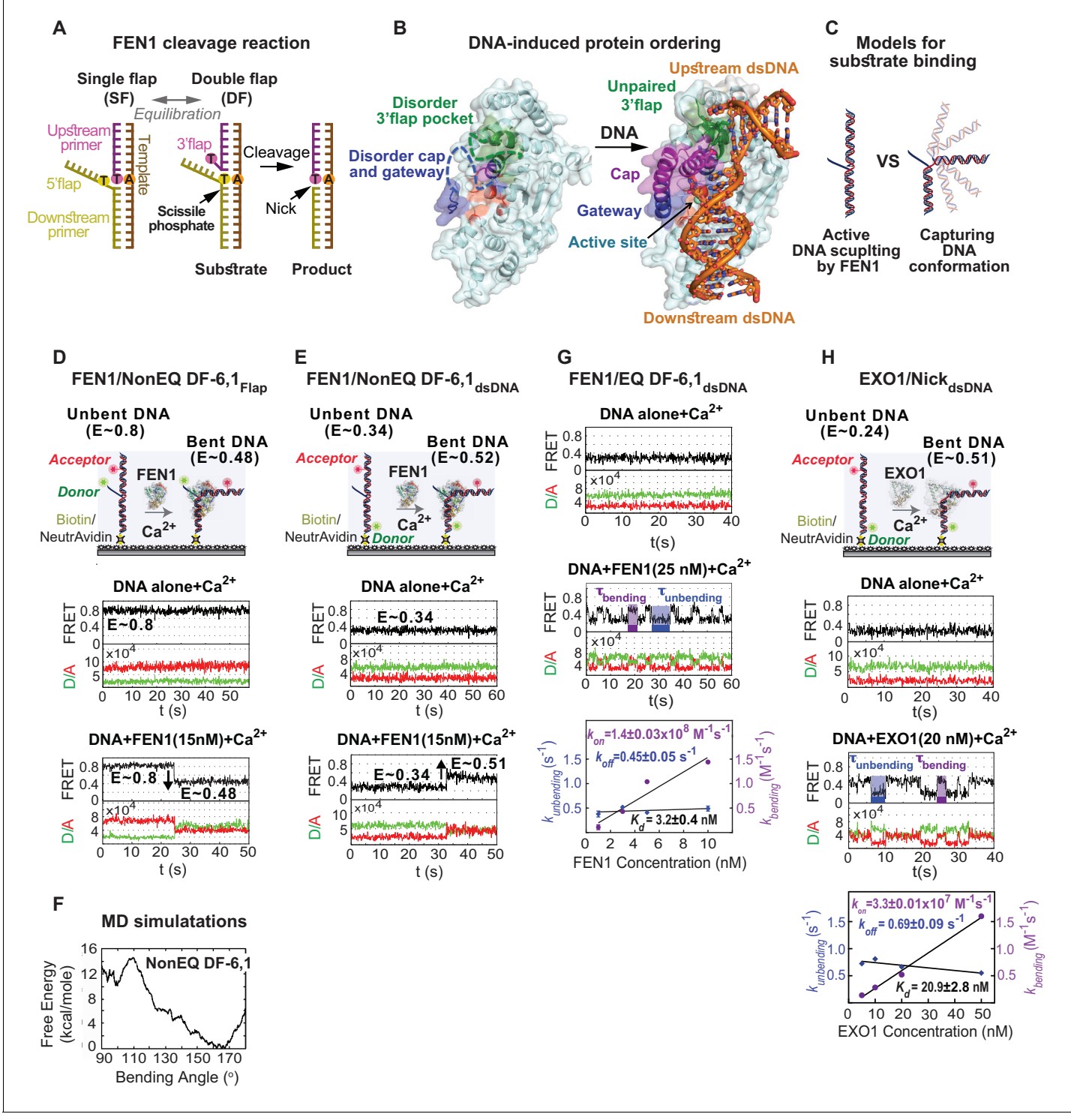

**Figure 1.** Active junction bending by structure-specific 5′nucleases. (**A**) FEN1 cleavage reaction. Schematic showing the equilibration of a flap substrate junction from a single- to a double-flap and its subsequent cleavage by FEN1 to generate a nick that can be sealed by DNA ligase 1. (**B**) Ordering of FEN1 upon DNA binding. FEN1 alone (1ULI.pdb) (*Sakurai et al., 2005*) and in complex with bent DNA (3Q8L.pdb) (*Tsutakawa et al., 2011*), highlighting the various structural features of FEN1 and the regions that undergo through disorder-to-order transitioning upon DNA binding. (**C**) Active DNA versus DNA conformational capturing models for forming the FEN1 complex with the bent DNA conformer. Monitoring DNA bending of FEN1 and non-equilibrated DF-6,1 using the flap-labeling scheme (NonEQ DF-6,1_Flap) (**D**) and internal labeling-scheme (NonEQ DF-6,1_dsDNA) (**E**). For each labeling, a schematic of the donor and acceptor positions (upper panel) and smFRET time traces of the substrate alone (middle panel) and in presence of FEN1 (lower panel) are shown; change in FRET upon DNA bending in each labeling scheme is highlighted. (**F**) Analysis of the structure of NonEQ

*Figure 1 continued on next page*

Figure 1 continued

DF-6,1 by MD simulations. The effective free energy profile (PMF) from adaptive biasing force calculations is shown. (G) Bending of equilibrated DF-6,1 (EQ DF-6,1$_{dsDNA}$) by FEN1. smFRET time traces of EQ DF-6,1$_{dsDNA}$ alone (upper panel) and in the presence of FEN1 (middle panel) and analysis of its DNA bending association rate constant ($k_{on-bending}$) and dissociation rate constant ($k_{off-unbending}$) (lower panel) are shown. $k_{bending}$ and $k_{unbending}$ were calculated by fitting an exponential function to the histogram from the population of dwell times of bent ($\tau_{bending}$) and unbent ($\tau_{unbending}$) conformers, respectively; error bars correspond to the standard deviation of the fit. $k_{on-bending}$ and $k_{off-unbending}$ are calculated from the slope of $k_{bending}$ from a linear regression fit and the mean of $k_{unbending}$, respectively; the error bars correspond to the standard deviation of the fit. $K_{d-bending} = k_{off-unbending}/k_{on-bending}$. (H) Bending of nicked substrate using the internal labeling scheme (Nick$_{dsDNA}$) by EXO1. A schematic of the donor and acceptor positions (upper panel), smFRET time traces of Nick$_{dsDNA}$ alone and in the presence of EXO1 (middle panels) and analysis of its $k_{on-bending}$, $k_{off-unbending}$ and $K_{d-bending}$ (lower panel) is presented. Donor and acceptor are at identical positions to those in DF-6,1$_{dsDNA}$ in *Figure 1E*. $k_{on-bending}$, $k_{off-unbending}$ and $K_{d-bending}$ were calculated as in 1G. All TIRF-smFRET experiments were acquired at 100 ms.

The following figure supplements are available for figure 1:

**Figure supplement 1.** Bending of equilibrated and non-equilibrated DF-6,1 by FEN1.

**Figure supplement 2.** Flap substrates exist as a stable extended conformer.

**Figure supplement 3.** MD simulations of the conformational states and DNA bending energy of nick and various flap structures.

**Figure supplement 4.** Bulk cleavage, SPR binding and time-resolved bulk FRET of selected substrates.

**Figure supplement 5.** Active bending of nicked DNA by EXO1.

---

but it is ordered when bound to the 3'flap (*Figure 1B*) (*Chapados et al., 2004*; *Tsutakawa et al., 2011*), suggesting DNA-induced ordering of the cap-helical gateway. Comparison of substrate and product complexes shows that the scissile phosphate nucleotide is fully paired at the nick junction and away from the active site's metal ions, suggesting that unpairing flanking nucleotides may move ssDNA into the active site (*Tsutakawa et al., 2011*).

By building upon these strong static structural and ensemble biochemical results, we reasoned that single-molecule experiments could resolve mechanistic unknowns by deconvoluting DNA bending, protein disorder-to-order transitioning, active-site assembly and incision. Like other 5'nucleases, FEN1 displays maximum catalytic efficiency for its cognate substrate but it is only residually active on substrates that vary only slightly (*Finger et al., 2009*; *Kao et al., 2002*). To define the mechanism for this catalytic efficiency and selectivity, we used single-molecule (sm)FRET at a millisecond to sub-millisecond temporal resolution to simultaneously measure in real time DNA conformational changes and catalysis when FEN1 encounters cognate or noncognate flap substrates as well as when the disorder-to-order transition or the active site is perturbed.

## Results

### FEN1 actively bends the DNA

A major question in DNA damage recognition is whether the DNA distortion observed in protein−DNA complexes occurs spontaneously and is captured by the protein (termed conformational selection) or if the protein actively sculpts the DNA into the distorted conformation (*Figure 1C*). To determine which is the case for FEN1, we started by establishing the conformational state of DF substrates alone using an ideal non-equilibrated (NonEQ) DF substrate containing 6 nt ssDNA 5'flap and 1 nt ssDNA 3'flap primers with no complementarity with the template strand (NonEQ DF-6,1). DNA bending was monitored by placing an Alexa Fluor-647 acceptor 12 nt into the upstream dsDNA and a Cy3 donor at the 5'flap end (NonEQ DF-6,1$_{Flap}$) (*Figure 1D*) or 15 nt into the downstream dsDNA (NonEQ DF-6,1$_{dsDNA}$) (*Figure 1E*). The substrates contained biotin to allow for immobilization on a polyethylene glycol-coated coverslip via biotin-NeutrAvidin linkage (*Figure 1D,E*). The experiments were performed using custom-built setups operating in either the total internal reflection fluorescence (TIRF) mode at a standard temporal resolution of 100 ms (*Sobhy et al., 2011*) as the primary method or the confocal mode for higher temporal resolution.

The single-molecule time traces of the substrate alone showed a single FRET state with no transition from this state (*Figure 1D,E*); FRET efficiency histograms generated from multiple single molecules fit to a single Gaussian (*Figure 1—figure supplement 1A,B*). The conformer of the duplex arms of the substrate was insensitive to variation in the concentration of divalent metal ions ($Mg^{2+}$) (*Figure 1—figure supplement 2A,B*), type of metal ion ($Mg^{2+}$ versus $Ca^{2+}$) (*Figure 1—figure supplement 1A,B* and *Figure 1—figure supplement 2A,B*) or 5'flap length (*Figure 1—figure supplement 2A,B*). The flap-labeling scheme was sensitive to variation in the $Mg^{2+}$ ion concentration when the 5'flap length exceeded 6 nt (*Figure 1—figure supplement 2B*). This could explain why previous work suggested that a double-flap substrate is a dynamic structure (*Craggs et al., 2014*). To test for short-lived alternative conformers, we used confocal-based smFRET to increase the temporal resolution to 5 ms on surface-immobilized DNA and to sub-ms on freely diffusing DNA in solution. Importantly, we found that substrate remained as a single conformer (*Figure 1—figure supplement 2C, D*). Potential mean force molecular dynamics (MD) simulations showed that extended DNA (~165°) was the most energetically favorable conformer in DF-6,1 (*Figure 1F* and *Figure 1—figure supplement 3A,B*). Base stacking at the nick junction and between the 3'flap base and the first base on the 5'flap stabilized this extended conformer (*Figure 1—figure supplement 3B*). The energetic cost required to bend the DNA up to ~140° was low (*Figure 1F*) and similar to that in dsDNA (*Sharma et al., 2013*). This was followed by a rapid increase in the energy required to surpass a significant barrier of ~14 kcal/mole to break the base stacking and bend the DNA (*Figure 1F*). These data suggest that DF substrate remains in an extended form that must be actively bent.

In fact, adding FEN1 to DNA NonEQ DF-6,1 in both labeling schemes showed transitions to the bent states in a single step to form a stable FEN1−DNA$_{bent}$ complex that rarely dissociated during our 60 s standard acquisition time (*Figure 1D,E*). We calculated the DNA bending dissociation constant ($K_{d-bending}$) from the FRET efficiency histogram-binding isotherm to be 3.9 ± 0.4 nM and 4.6 ± 0.6 nM for NonEQ DF-6,1$_{Flap}$ and NonEQ DF-6,1$_{dsDNA}$, respectively (*Figure 1—figure supplement 1A,B*). This dissociation constant agreed with the nM range of $K_m$ from bulk cleavage assays (*Figure 1—figure supplement 4A*) (*Finger et al., 2009*) and the DNA binding dissociation constant ($K_{d-binding}$) of FEN1 and NonEQ DF-6,1 as determined by surface plasmon resonance (SPR) (*Figure 1—figure supplement 4B*). The change in FRET in both NonEQ DF-6,1$_{Flap}$ and NonEQ DF-6,1$_{dsDNA}$ was confirmed by time-resolved bulk FRET measurements (*Figure 1—figure supplement 4C*).

To mimic the in vivo junction, we used equilibrated (EQ) DF-6,1 (*Figure 1A*). FEN1 actively bent EQ DF-6,1$_{dsDNA}$ to a similar extent and similar $K_{d-bending}$ as DF-NonEQ 6,1$_{dsDNA}$ (*Figure 1—figure supplement 1B,C*). Nonetheless, time traces showed multiple transitions between bent and unbent states (*Figure 1G*). The reduced stability of the bent conformer in the equilibrated substrate suggests that a bound 3'flap could dissociate from the 3'flap-binding pocket. The dissociated 3'flap in the equilibrated substrate would pair with the template strand before FEN1 could rebind it while in the non-equilibrated substrate it would remain available for rebinding FEN1. Dwell time analysis of the bent ($\tau_{bending}$) and unbent ($\tau_{unbending}$) states at increasing FEN1 concentrations indicated that the apparent first-order rate constant for DNA bending ($k_{bending} = 1/\tau_{bending}$) increased linearly while that for DNA unbending ($k_{unbending} = 1/\tau_{unbending}$) remained constant (*Figure 1G*). This is the trend expected for a 1:1 binding equilibrium where $k_{bending}$ and $k_{unbending}$ correspond to the association and dissociation of FEN1, respectively. Notably, the second-order association rate constant ($k_{on-bending}$) calculated from the slope of the linear fit of the concentration dependence of $k_{bending}$ was diffusion-limited (1.4 ± 0.03×$10^8$ $M^{-1}$ $s^{-1}$), and the average value of $k_{unbending}$ ($k_{off-unbending}$) was 0.45 ± 0.05 $s^{-1}$ (*Figure 1G*).

It is unclear what caused the much higher $K_{d-bending}$ reported in our earlier work (*Sobhy et al., 2013*), but we suggest that both slower association and faster dissociation rates were influenced. Nonetheless, the FRET states of NonEQ DF-6,1$_{Flap}$ alone and when bent by FEN1 and the relative comparison of bending the cognate with the noncognate substrates are similar under low and high $K_{d-bending}$ conditions as shown below.

To see if active bending of the ss/ds-DNA junction may be a conserved feature in 5'nucleases, we tested human mismatch repair exonuclease 1 (EXO1), which recognizes an ideal junction of either a nick or a 3' overhang (*Orans et al., 2011*). EXO1 actively bent a DNA nick with diffusion-limited $k_{on-bending}$ (*Figure 1H*); the donor and acceptor had identical positions to those in NonEQ DF-6,1$_{dsDNA}$. Free MD simulations showed that the nick behaved similarly to flap substrates for bending angles in

the 140°−180° range (*Figure 1—figure supplement 3C*). The bent conformer of EXO1 had similar FRET to that of FEN1 (*Figure 1—figure supplement 1B* and *Figure 1—figure supplement 5*), consistent with the structures of their DNA complexes (*Orans et al., 2011*; *Tsutakawa et al., 2011*).

## FEN1 never misses cleavage of its correct substrate

To examine active-site assembly with respect to DNA bending, we replaced $Ca^{2+}$ with $Mg^{2+}$ to simultaneously monitor DNA bending and 5'flap cleavage using the flap-labeling scheme (*Figure 2A*). Time traces indicated that FEN1 always bent NonEQ DF-6,1$_{Flap}$ before cleaving the Cy3-containing 5'flap; remarkably every DNA bending event led to a successful cleavage reaction (*Figure 2A*, *Figure 2—figure supplement 1A*). We confirmed DNA bending before cleavage by the

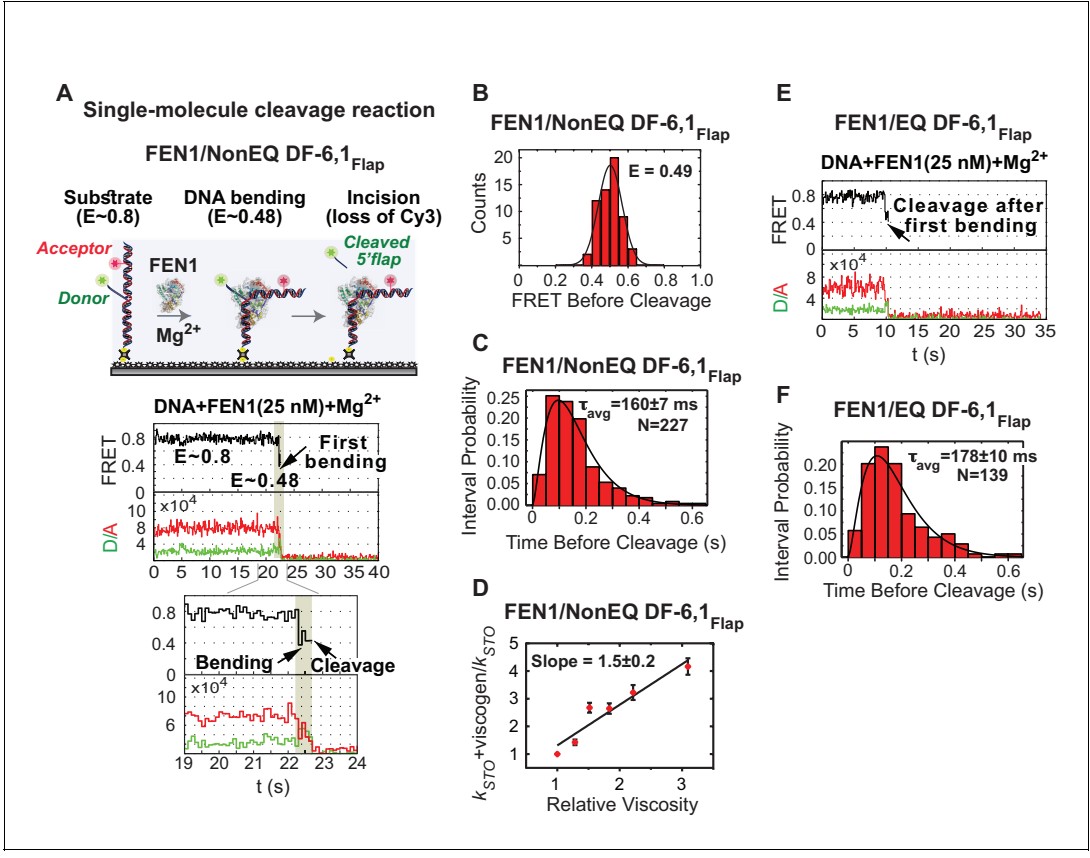

**Figure 2.** Cleavage of cognate substrate by FEN1. (A) Cleavage of NonEQ DF-6,1$_{Flap}$. Schematic showing the simultaneous monitoring of DNA bending and 5'flap cleavage at the single-molecule level (upper panel). A representative smFRET time trace with a zoomed-in view showing the cleavage of NonEQ DF-6,1$_{Flap}$ in which FEN1 never misses the opportunity to bend the DNA and cleave it (lower panel). (B) FRET of the bent state before cleavage of NonEQ DF-6,1$_{Flap}$ fitted with a Gaussian distribution from multiple cleavage events. (C) Dwell times of the bent state prior to cleavage of NonEQ DF-6,1$_{Flap}$ fitted with a gamma distribution to calculate average dwell time ($\tau_{avg}$) from the number of independent experiment N = 3; the uncertainty corresponds to the standard error of the mean. Single turnover $k_{cat}$ ($k_{STO}$) is = 1/$\tau_{avg}$. Cleavage was performed at 50 ms temporal resolution. (D) Effect of low molecular weight viscogen (glycerol) on disorder-to-order transitioning in FEN1. Graph showing relative $k_{STO}$ of NonEQ DF6,1$_{Flap}$ cleavage upon addition of glycerol at increasing relative viscosity from N = 2–3 fitted with a linear regression to calculate the slope of the curve; the error corresponds to the standard error of the mean. $k_{STO}$ was determined as in *Figure 2C*. (E) A representative smFRET time trace showing the cleavage of EQ DF-6,1$_{Flap}$ in which FEN1 never misses the opportunity to bend the DNA and cleave it. (F) A histogram showing the distribution of dwell times of the bent state prior to cleavage of EQ DF6,1$_{Flap}$. $\tau_{avg}$ from N = 3 is calculated as in *Figure 2C* at a temporal resolution of 50 ms.

The following figure supplements are available for figure 2:

**Figure supplement 1.** Single-molecule cleavage of cognate substrates and probing conformational changes to FEN1 by viscogens.

**Figure supplement 2.** Controls for assigning donor loss to the cleavage and immediate departure of the 5'flap.

clear anti-correlated change in the donor and acceptor intensities (*Figure 2A*). Direct comparison of donor fluorescence in the presence of FEN1 and either $Mg^{2+}$ or $Ca^{2+}$ ions indicated that there is a strong correlation between the loss of donor particles and the presence of $Mg^{2+}$ that coincided with the introduction of FEN1 into the flow cell (*Figure 2—figure supplement 2A,B*). This confirms that the loss of donor particles is due to 5'flap cleavage and not due to donor photobleaching. Analysis of FRET values before cleavage from individual time traces showed that FEN1 cleaved NonEQ DF-$6,1_{Flap}$ from a fully bent state (*Figure 2B*) and remained bent for $160 \pm 7$ ms prior to cleavage (*Figure 2C*). FEN1 cleavage generates two products: 5'flap ssDNA and nicked dsDNA (*Figure 1A*). Previous studies demonstrated that excess nicked dsDNA but not 5'flap ssDNA influences FEN1 activity, which suggests that only nicked dsDNA is a competitive inhibitor of FEN1 release (*Finger et al., 2009*; *Tarantino et al., 2015*). Consistent with these findings, we also observed that the lag time before cleavage is not influenced by the presence of excess 5'flap ssDNA (*Figure 2—figure supplement 2C*). Furthermore, SPR showed only residual transient binding of FEN1 to ssDNA at concentrations that were orders of magnitude above $K_{d-bending}$ of DF-6,1 (*Figure 2—figure supplement 2D*). Our single-molecule cleavage measurement is not inhibited by lack of 5'flap ssDNA product release. Therefore single-turnover $k_{cat}$ ($k_{STO}$) could be determined directly from the lag time prior to cleavage ($k_{STO} = 1/\tau_{before\ cleavage}$). However, since 5'flap release would still contribute to the dwell time before cleavage, our single turnover should be treated as an apparent value. Notably, our $k_{STO}$ ($6.3 \pm 0.2$ s$^{-1}$) was comparable to that determined by bulk cleavage assays ($k_{STO}$ 12.3 s$^{-1}$−>5 s$^{-1}$) (*Algasaier et al., 2016*; *Stodola and Burgers, 2016*), with the slight difference explained by the lower reaction temperature in the single molecule assays. The diffusion-limited rates of DNA bending and cleavage before protein dissociation provide direct evidence that the reaction of FEN1 on a cognate substrate is limited by encounters between the enzyme and the substrate.

The lag time distribution prior to cleavage shows a rise and decay (*Figure 2C*), suggesting that the underlying catalytic mechanism after the diffusion-limited DNA bending step involves two or more steps, as a single-step process will have a single exponential decay. We reasoned that these steps likely include 3'flap-induced disorder-to-order transitioning and cleavage chemistry. To test this idea, we employed glycerol and sucrose as low-molecular-weight viscogen to slow any local protein conformational change that mediates catalysis and/or product release. Increasing glycerol concentration decreased $k_{STO}$ linearly with a slope of $1.5 \pm 0.2$ (*Figure 2D* and *Figure 2—figure supplement 1B*); a similar effect was observed with sucrose (*Figure 2—figure supplement 1C*). Yet, $k_{STO}$ was unaffected by polyethylene glycol-8000, a high-molecular-weight viscogen that is too large to interfere with local protein conformational changes (*Figure 2—figure supplement 1D*). $k_{STO}$ is not influenced by 5'flap ssDNA product release, suggesting that 3'flap-induced protein ordering is a terminal step to verify the substrate before incision. The shape of the histograms in the presence of viscogen however remains a rise and decay (*Figure 2—figure supplement 1B*), in contrast to the prediction of collapsing into a single exponential decay should protein-ordering acts in a single rate-limiting step. This suggests that the 3'flap-induced protein ordering is likely involves multistep processes that are being slowed down by the presence of viscogen and/or these multisteps control different rate-limiting steps during catalysis such as DNA unpairing and/or DNA shifting into the active site. Biologically relevant, the cleavage behavior from the first DNA bending and the $k_{STO}$ were similar whether there was a deliberate mispaired 3'flap (NonEQ DF-6,1; *Figure 2A,C* and *Figure 2—figure supplement 1A*) or an equilibrating 3'flap (EQ DF-6,1; *Figure 2E,F* and *Figure 2—figure supplement 1E*), consistent with bulk cleavage reactions (*Tarantino et al., 2015*).

## FEN1 actively creates a 3'flap in equilibrated DF substrate

From bulk measurements, it remains unclear how 3'flap-induced protein ordering operates in the case of the in vivo equilibrated DF substrate. The equilibrated junction may exist as a single 5'flap that requires active molding by FEN1 into a double 5'- and 3'-flap or as a DF with a readily available 3'flap for FEN1 capturing. To address this, we started by investigating the requirement of having a preformed 3'flap for inducing DNA bending. Removal of the 3'flap from NonEQ DF while maintaining its 5'flap (a substrate termed single flap (SF)) decreased FEN1 cleavage activity by 34 fold (*Finger et al., 2009*). Time traces on surface-immobilized SF-$6,0_{Flap}$ accessed at 5 ms using confocal-based smFRET showed that FEN1 actively bent SF (*Figure 3A*). The $\tau_{bending}$ was markedly reduced to ~43 ms ($k_{off-unbending} = 23.3 \pm 3.8$ s$^{-1}$) in contrast to that of the stable bent conformer in NonEQ

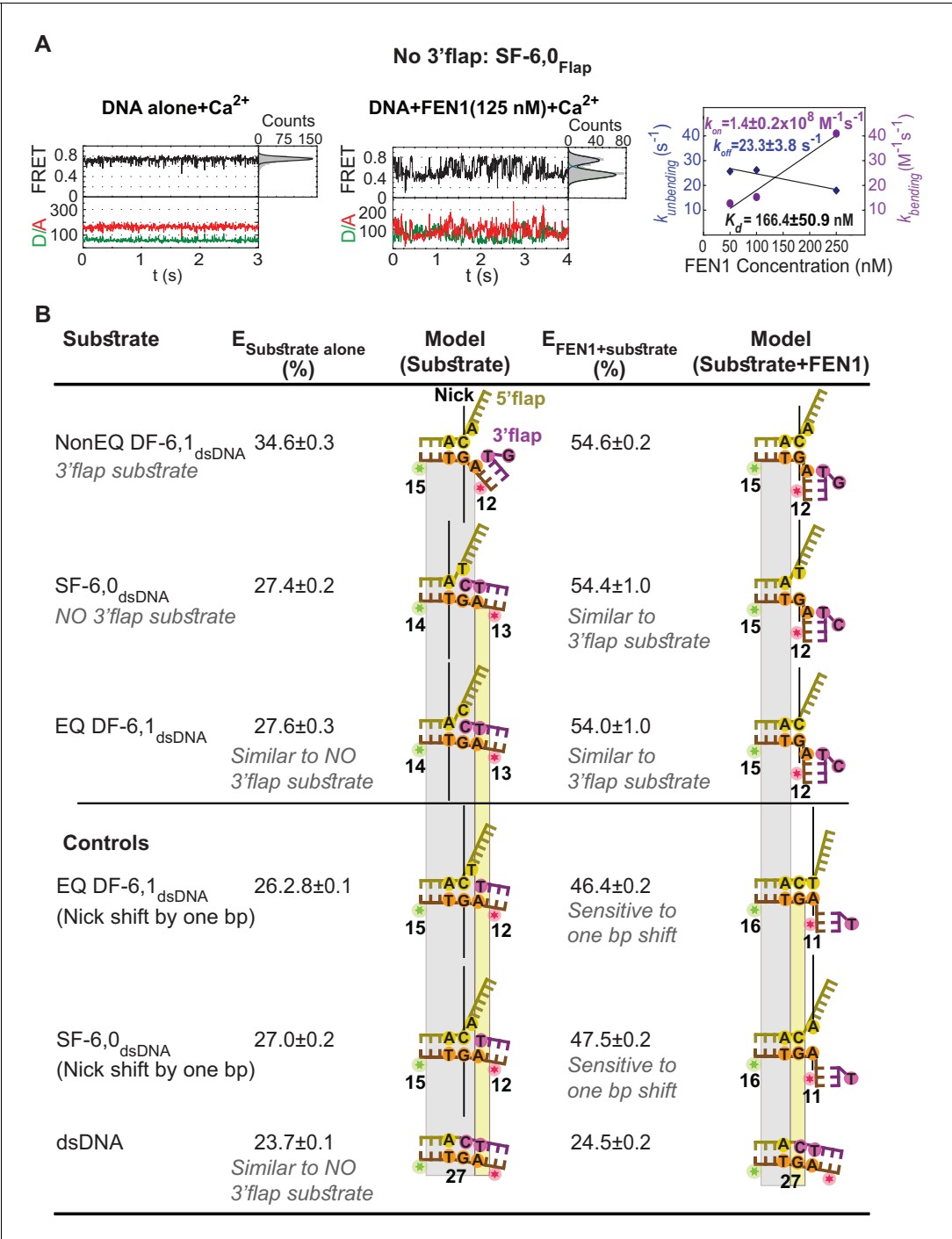

**Figure 3.** Active sculpting of the 3'end of a nick junction creates a 3'flap and drive protein ordering. (**A**) Confocal-based smFRET time traces of surface-immobilized SF6,0$_{Flap}$ alone (left panel) and in the presence of FEN1 (middle panel) acquired at 5 ms, showing rapid transitions from high to low FRET upon DNA bending. $k_{on-bending}$ and $k_{off-unbending}$ of SF6,0$_{Flap}$ by FEN1 (right panel) calculated as in *Figure 1G*. (**B**) FEN1 actively creates a 3'flap at the nick junction of cognate and noncognate substrates. Determining the status of the 3'flap in equilibrated DF and SF junctions by comparing the FRET states of various nick-junction positions in NonEQ DF6,1$_{dsDNA}$, EQ DF6,1$_{dsDNA}$ and SF6,0$_{dsDNA}$ in the absence or presence of FEN1. 0.5 nM DNA and saturating concentrations of FEN1 were used (1000 nM for SF6,0$_{dsDNA}$ and its one base pair shift construct and 200 nM for the remaining constructs). FRET values were determined by fitting the burst confocal-smFRET histograms from freely diffusing DNA at sub-ms temporal resolution with a Gaussian. FRET is reported as a percentage, and the uncertainty corresponds to the standard deviation of N = 3.

The following figure supplement is available for figure 3:

*Figure 3 continued on next page*

*Figure 3 continued*

**Figure supplement 1.** All histograms corresponding to the data shown in *Figure 3B*.

DF-6,1 (*Figure 3A*). However, the $k_{on\text{-}bending}$ remained limited by diffusion and similar to that of EQ DF-6,1$_{dsDNA}$ (*Figure 3A*). $K_{d\text{-}bending}$ was 50-fold higher than for NonEQ DF-6,1 (*Figure 3A*), consistent with that observed from confocal-based smFRET with burst analysis of freely diffusing SF-6,0$_{dsDNA}$ (*Figure 1—figure supplement 4D*) and the markedly increased $K_{d\text{-}binding}$ by SPR (*Figure 1—figure supplement 4B*). These results show that a 3'flap is not required for DNA bending but it is critical for DNA binding stability.

We next established whether the equilibrated junction existed as a SF or DF by comparing the FRET states of substrates containing only 5'flap, nonequilibrated 5'- and 3'-flap, or equilibrated 5'- and 3'-flap. Interestingly, we found that the FRET values of the substrate alone in EQ DF-6,1$_{dsDNA}$ and in SF-6,0$_{dsDNA}$ were similar (E ~ 0.27), but less than that in NonEQ DF-6,1$_{dsDNA}$ (E ~ 0.34) (*Figure 3B*; *Figure 3—figure supplement 1*). The geometry of the equilibrated DF and SF was slightly less extended than that of dsDNA (E ~ 0.23) (*Figure 3B*; *Figure 3—figure supplement 1*). This suggests that the equilibrated junction existed as a SF, which was shown by free MD simulations not to equilibrate to a DF (*Figure 1—figure supplement 3D,E*).

The finding that FEN1 cleaves equilibrated and non-equilibrated DF substrates with equal activity (*Tarantino et al., 2015*) prompted us to propose that FEN1 actively converts the equilibrated junction from a SF to a DF by pulling out its 3'end to create a 3'flap. The nick position relative to the donor and acceptor would differ by one base pair if EQ DF-6,1$_{dsDNA}$ existed in a SF form compared to that in NonEQ DF-6,1$_{dsDNA}$ (*Figure 3B*; *Figure 3—figure supplement 1*). Creating a 3'flap would thus move the junction back to the identical position of that in NonEQ DF-6,1$_{dsDNA}$ (*Figure 3B*; *Figure 3—figure supplement 1*). In a control experiment, we showed that we were able to detect junction movement by one base pair because a deliberate difference of one base pair in the nick position of EQ DF-6,1$_{dsDNA}$ would result in a detectable difference in the FRET value of their bent conformers (E ~ 0.46 versus E ~ 0.54) (*Figure 3B*; *Figure 3—figure supplement 1*). The bent conformer in EQ DF-6,1$_{dsDNA}$ had identical FRET to that in NonEQ DF-6,1 (E ~ 0.54) (*Figure 3B*; *Figure 3—figure supplement 1*), demonstrating that the equilibrated nick junction must have moved by one base pair. These data therefore suggest that FEN1 actively sculpts the 3'end of its in vivo equilibrated nick junction to create a 3'flap and to drive its ordering. The active DNA bending and its subsequent use to induce protein ordering through active formation of a 3'flap suggest that there is an induced-fit mechanism between FEN1 and DNA that functions in a mutual way.

## Substrate verification by 3'flap-induced protein ordering

The ability of FEN1 to actively create a 3'flap at the nick junction of its equilibrated DF substrate raises the possibility that it could also create a 3'flap at nick junctions of noncognate substrates. This mechanism would explain why the cleavage site is shifted by 1 nt in SF versus DF substrates (*Finger et al., 2009*). We found that the FRET value of the bent conformer in SF-6,0$_{dsDNA}$ was also similar to that of NonEQ DF-6,1$_{dsDNA}$ (*Figure 3B*; *Figure 3—figure supplement 1*), demonstrating that the nick junction must have moved by one base pair. In a control experiment we showed that a shift of one base pair in the nick position in SF-6,0$_{dsDNA}$ resulted in a detectable difference in the FRET of the bent conformer as observed in the case of EQ DF-6,1$_{dsDNA}$ (*Figure 3B*; *Figure 3—figure supplement 1*). This indicates that FEN1 creates 3'flaps at noncognate nick junctions, suggesting that there is another requirement during substrate validation. In an in vivo-equilibrating junction, the nick structure would be maintained, while in noncognate substrates, a one-nucleotide mismatch would be added at the junction (*Figure 3B*; *Figure 3—figure supplement 1*). FEN1 discriminates against such a structure with 33-fold reduced activity (*Beddows et al., 2012*). Here, the $K_{d\text{-}bending}$ of DF-6,1 containing a one-nucleotide mismatch at the junction (termed DF-7,1$_{mismatch(1nt)\text{-}Flap}$) increased by seven fold (*Figure 4A*), with the time traces showing a less-stable bent conformer (*Figure 4B* and *Figure 4—figure supplement 1A*). Since FEN1 forms 3'flaps for both cognate and noncognate substrates, only the junctions that are fully paired are therefore stably bent.

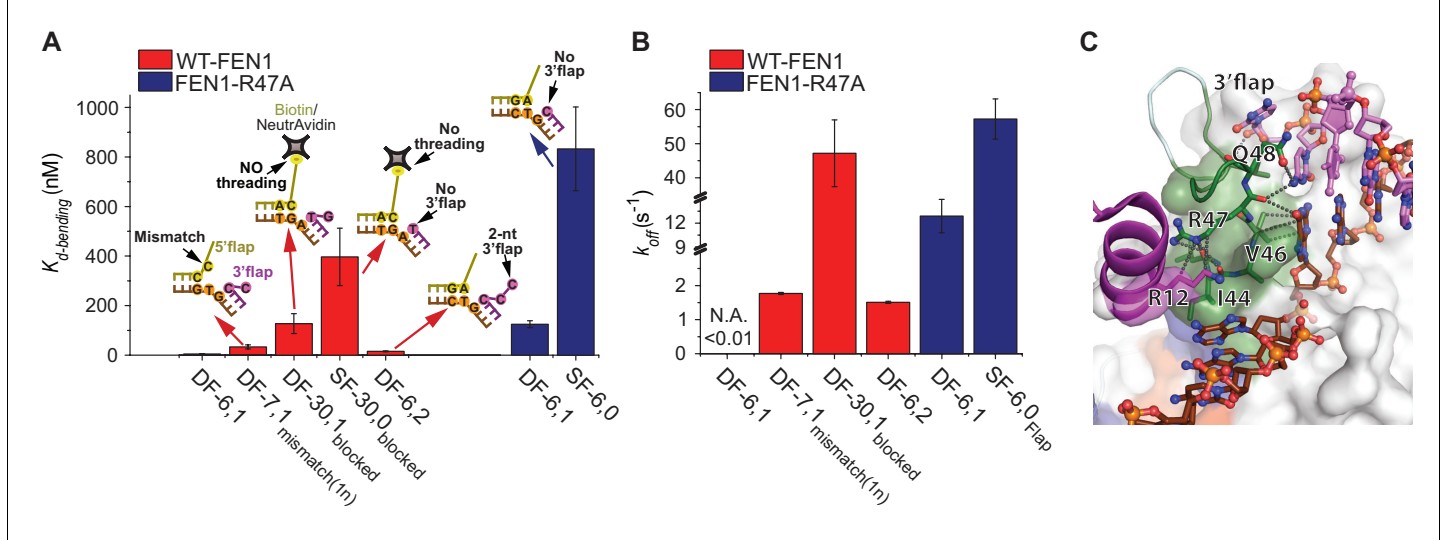

**Figure 4.** Verification of the bent DNA conformer by the 3'flap-induced protein ordering. (**A**) Bar chart comparing $K_{d\text{-}bending}$ for FEN1-WT or FEN1-R47A on various non-equilibrating flap substrates using the internal labeling scheme. Used noncognate substrates include SF-6,0, DF containing 1 nt mismatch at the nick junction (DF-7,1$_{mismatch(1nt)}$), DF containing biotin-NeutrAvidin on the 5'flap to block 5'flap threading (DF-30,1$_{blocked}$) and its SF version (SF-30,0$_{blocked}$), and DF containing 2 nt 3'flap (DF-6,2). (**B**) Bar chart comparing $k_{off\text{-}unbending}$ for FEN1-WT or FEN1-R47A on various non-equilibrating flap substrates using the internal labeling scheme. The lower estimate of $k_{off\text{-}unbending}$ for FEN1-WT on DF-6,1 corresponds to the 60 s acquisition time where transitions were rarely detected. $K_{d\text{-}bending}$ and $k_{off\text{-}unbending}$ are calculated as in *Figure 1—figure supplement 1A* and *Figure 1G*, respectively. $k_{off\text{-}unbending}$ was determined from multiple FEN1 concentrations except for FEN1-R47A on SF-6,0 and FEN1 on DF-7,1$_{mismatch(1nt)}$, which were determined from two and one concentration, respectively. The smFRET technique and temporal resolutions used in *Figure 4A,B* are described in *Figure 4—figure supplement 1*. (**C**) R47 acts as a sensor that couples structuring of the 3'flap-binding pocket and the cap-helical gateway. R47 in the hydrophobic wedge mediates multiple interactions, where it stacks against the first base pair on the 3'flap side of the junction while its side chain C-caps the α2 in the gateway (highlighted in green) and stacks with K128 on α5 in the cap (highlighted in purple) (3Q8L.pdb) (*Tsutakawa et al., 2011*).

The following figure supplement is available for figure 4:

**Figure supplement 1.** Bending kinetics of various noncognate substrates.

The combined requirement for 3'flap and base pairing at the junction suggests that signaling occurs via ordering from the 3'flap-binding pocket to the distant gateway where the 5'flap is recognized and cleaved. The superfamily semi-conserved R47 in the hydrophobic wedge is poised to mediate this coordination: it stacks against the first base pair on the 3'flap side of the junction while its side chain C-caps the α2 in the gateway and stacks with K128 on α5 in the cap (*Tsutakawa et al., 2011*) (*Figure 4C*). Mutating R47 to A (FEN1-R47A) disabled FEN1's cleavage on DF substrate to a similar extent as wild-type (wt)-FEN1's cleavage on SF-6,0 (*Tsutakawa et al., 2011*). To test this allosteric signaling idea, we maintained the 3'flap binding using NonEQ DF-6,1$_{dsDNA}$ while altering R47 using FEN1-R47A. The defects in $K_{d\text{-}bending}$ and $k_{off\text{-}unbending}$ resemble those in wt-FEN1 on SF-6,0 (*Figure 4A,B* and *Figure 4—figure supplement 1B*). We determined that FEN1-R47A engaged the 3'flap because $K_{d\text{-}bending}$ and $k_{off\text{-}unbending}$ increased when SF-6,0$_{dsDNA}$ was used rather than NonEQ DF-6,1$_{dsDNA}$ (*Figure 4A,B*).

We next investigated the communication between the 3'flap-induced protein ordering and the distant gateway with respect to 5'flap recognition. It has been postulated that the 5'flap may thread through the cap-helical gateway and that this threading is needed for catalysis (*Gloor et al., 2010*; *Patel et al., 2012*; *Sobhy et al., 2013*) or that the 5'flap may be clamped away from the active site for catalysis (*Orans et al., 2011*). To test possible threading and its coordination with the 3'flap-induced protein ordering, we used a modification that prevented 5'flap threading. Blocking the threading by immobilizing a DF substrate through a biotin attached at the end of a 30 nt ssDNA 5'flap (termed DF-30,1$_{blocked\text{-}dsDNA}$) impaired DNA bending to comparable $K_{d\text{-}bending}$ and $k_{off\text{-}unbending}$ of SF-6,0 (*Figure 4A,B* and *Figure 4—figure supplement 1C*), consistent with the markedly

reduced $K_{d\text{-}binding}$ captured by SPR (*Figure 1—figure supplement 4B*). Notably, the bent DNA in the unthreaded substrate was distorted, but it did not reach the same final FRET state as when the 5'flap was not blocked (*Figure 4—figure supplement 1D*). This significant DNA distortion was masked in our previous experiment that relied on a flap labeling scheme to infer to the geometry of the blocked-threaded complex (*Sobhy et al., 2013*). Importantly, our new results indicate that initial DNA bending by FEN1 did not require threading, but that full bending required 5'flaps, if present, to be able to thread. The increased $K_{d\text{-}bending}$ of the unthreaded substrate upon removal of its 3'flap (SF-30,0$_{blocked\text{-}dsDNA}$) (*Figure 4A*) indicates that 3'flap binding did not require 5'flap threading. However, the ability of the 5'flap to thread is required for the 3'flap-induced protein ordering to form the stably and correctly bent DNA conformer.

Collectively, these results demonstrate that FEN1 bends both cognate and noncognate substrates and that $K_{d\text{-}bending}$ is higher for noncognate substrates. This is consistent with our previous findings under high $K_{d\text{-}bending}$ conditions (*Sobhy et al., 2013*). They further showed that FEN1 stabilizes the cognate substrate through remarkable selectivity for its key features of a fully paired nick junction, a 3'flap and a 5'flap while promoting the dissociation of noncognate substrates. Our observation of FEN1's ability to significantly bend the DNA in the blocked-threaded complex challenges our previous conclusion that 5'flap threading is strictly required to induce DNA bending (*Sobhy et al., 2013*). Our new results are consistent with a model in which FEN1 actively bends DNA to interact with the ss/ds-DNA junctions and subsequently verifies these interactions by the 3'flap-induced protein ordering.

## FEN1 avoids off-target DNA cleavage in the DNA lockdown step

To test the mechanism for assembly of catalytically competent active sites for cognate substrate incision, we compared the lifetime of the bent conformer to the lag time for cleaving correct versus incorrect substrates. In SF-6,0, the lifetime of the bent conformer was ~3.5 fold shorter than the required lag time prior to cleaving the cognate substrate (*Figure 3A* and *Figure 2C*). Traces of single-molecule cleavage showed that SF-6,0 underwent multiple cycles of DNA bending and unbending before a successful DNA bending event led to 5'flap cleavage (*Figure 5A*; *Figure 5—figure supplement 1A*; *Figure 2—figure supplement 2E*). These abortive DNA bending events are masked in bulk cleavage assays, which leads to underestimation of both $k_{STO}$ and the accuracy of FEN1 cleavage. Following the FEN1 cleavage reaction at the single-molecule level clearly leads to additional information. Similar results were observed in the cleavage of DF-6,1 by FEN1-R47A (*Figure 5B* and *Figure 5—figure supplement 1B*). These results show that destabilizing the bent DNA intermediate to rates that are limiting for catalysis reduces the probability of assembling catalytically competent active sites.

Interestingly, we also observed similar bending behavior without cleavage of FEN1 on noncognate substrates under conditions in which the lifetime of the bent conformer does not limit catalysis. DF-7,1$_{mismatch(1nt)\text{-}Flap}$ and DF containing 2 nt 3'flap (DF-6,2$_{Flap}$) exhibited $k_{off\text{-}unbending}$ that was ~13–15 fold slower than that of SF-6,0 and ~3–4 fold longer than $k_{STO}$ of the cognate substrate (*Figure 4B* and *Figure 2C*), yet FEN1 still bent these substrates multiple times without cleaving them (*Figure 5C,D* and *Figure 5—figure supplement 1C,D*). FEN1 therefore likely has intrinsic mechanisms that block the probable formation of catalytically competent active sites in noncognate substrates to inhibit off-target incision.

We reasoned that there are two possible mechanisms for controlling incision that can be tested experimentally. The 3'flap-induced protein ordering could act once per DNA bending event, locking the DNA into either a catalytically competent or incompetent conformation. In this mechanism, the $k_{STO}$ after DNA bending should be similar between cognate and noncognate substrates regardless of whether or not the lifetime of the bent conformer is limiting. Alternatively, the protein could lock the DNA into a bent conformer and go through multiple cycles of disorder-to-order transitioning to search for a catalytically competent conformation of protein and DNA. In this mechanism, the $k_{STO}$ would be slower for noncognate substrates, particularly under conditions when the lifetime of the bent conformer exceeded that required for cleavage. We found that the $k_{STO}$ of FEN1 in all tested noncognate substrates was similar and comparable to that in the cognate substrate (*Figure 5A–D* and *Figure 2C*). This indicates that the 3'flap-induced protein ordering locked the DNA into either a catalytically competent conformation to be immediately incised or into an incompetent conformation

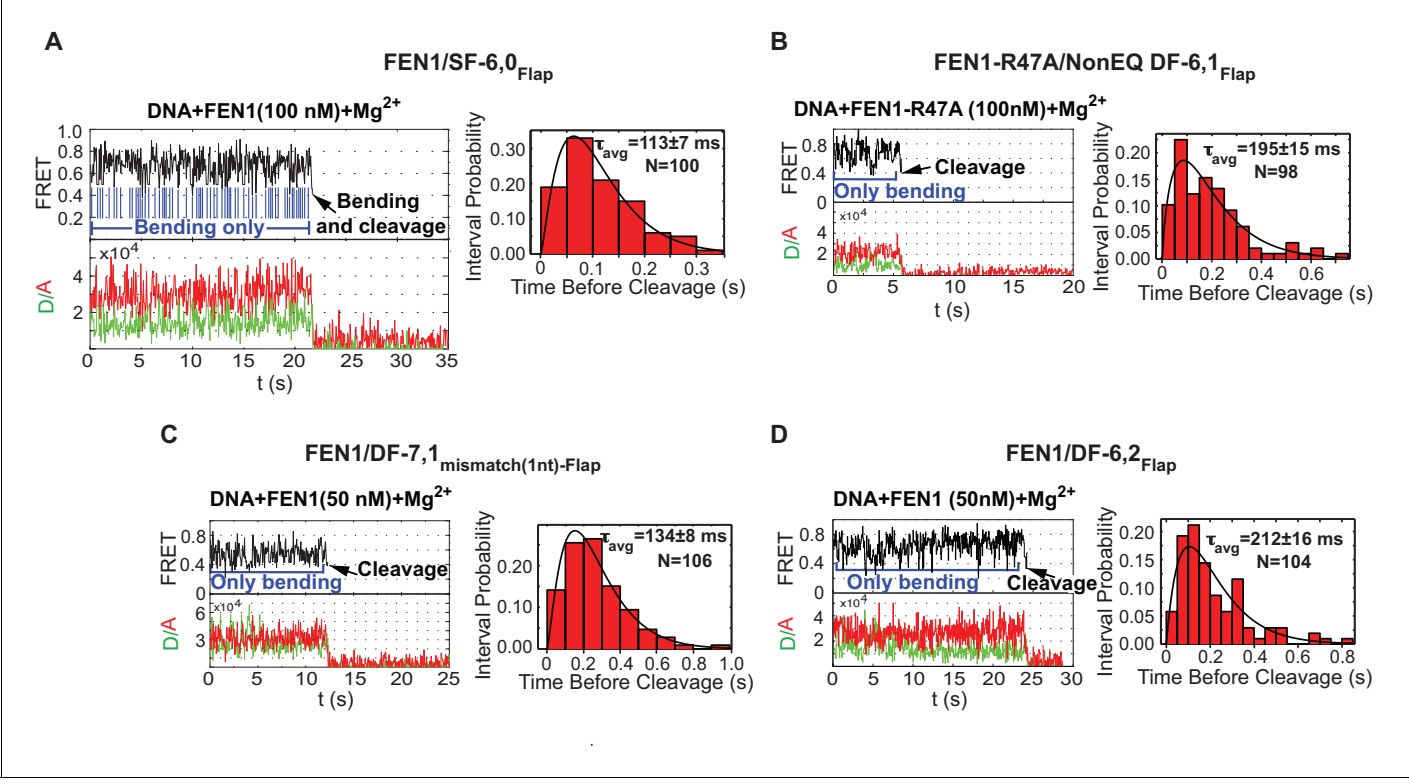

**Figure 5.** Cleavage of noncognate substrates by FEN1. (**A**, **B**, **C** and **D**) Representative smFRET time traces showing the cleavage of FEN1 on SF6,0$_{Flap}$, FEN1-R47A on NonEQ DF6,1$_{Flap}$, FEN1 on DF-7,1$_{mismatch(1nt)-Flap}$ and FEN1 on DF-6,2$_{Flap}$, respectively and the distribution of dwell times of the bent state prior to cleavage ($\tau_{avg}$). Cleavage occurs after a random number of missed cleavage opportunities from a bent conformer as illustrated in blue arrows in **Figure 5A**. $\tau_{avg}$ is calculated from N = 4 as in **Figure 2C** at a temporal resolution of 50 ms.

The following figure supplement is available for figure 5:

**Figure supplement 1.** FEN1 cleavage of various noncognate substrates.

that led to immediate DNA release from the bent conformation. Directly observing FEN1 conformation will lead to further understanding on how it prevents cleavage in noncognate substrates.

## Role of active site residues in positioning the 5'flap and junction

Given that FEN1 actively bends the DNA, it is unclear how FEN1 positions the junction before it locks down the DNA conformation. We therefore tested the role of key active-site residues. Mutating individual gateway residues Y40, K93, R100 or one of the metal-coordinating aspartic acidic residues (D181) to alanine markedly reduced the bent conformer's stability (**Figure 6A**). These results revealed a direct role for active-site residues in stabilizing the bent DNA conformer. Interestingly, $k_{on-bending}$ was reduced up to 11 fold in gateway mutants, with R100 and to a lesser extent Y40 being critical residues (**Figure 6B**). This result is surprising, given that these gateway residues appear to be disordered prior to DNA binding. Thus, active-site residues evidently contribute to active positioning of the junction while the 5'flap is being threaded through the unstructured cap-helical gateway. Moreover, D181A had only a minor effect on $k_{on-bending}$, implying that the metal ions do not interact with the phosphates during DNA bending and 5'flap threading. Importantly, this could provide a mechanism that protects the 5'flap from nonspecific cleavage.

FEN1-Y40A is the only active-site mutation that retains activity, albeit with a 100-fold reduced $k_{STO}$ (**Algasaier et al., 2016**). Y40 is proposed to play various roles in substrate positioning, including placing FEN1 at the junction, at the 5'flap in the cap-helical gateway and at the scissile phosphate in the active site (**Algasaier et al., 2016**; **Tsutakawa et al., 2011**). We observed that FEN1-Y40A cleaves DF-6,1 after multiple cycles of DNA bending and unbending (**Figure 6C**, **Figure 5—**

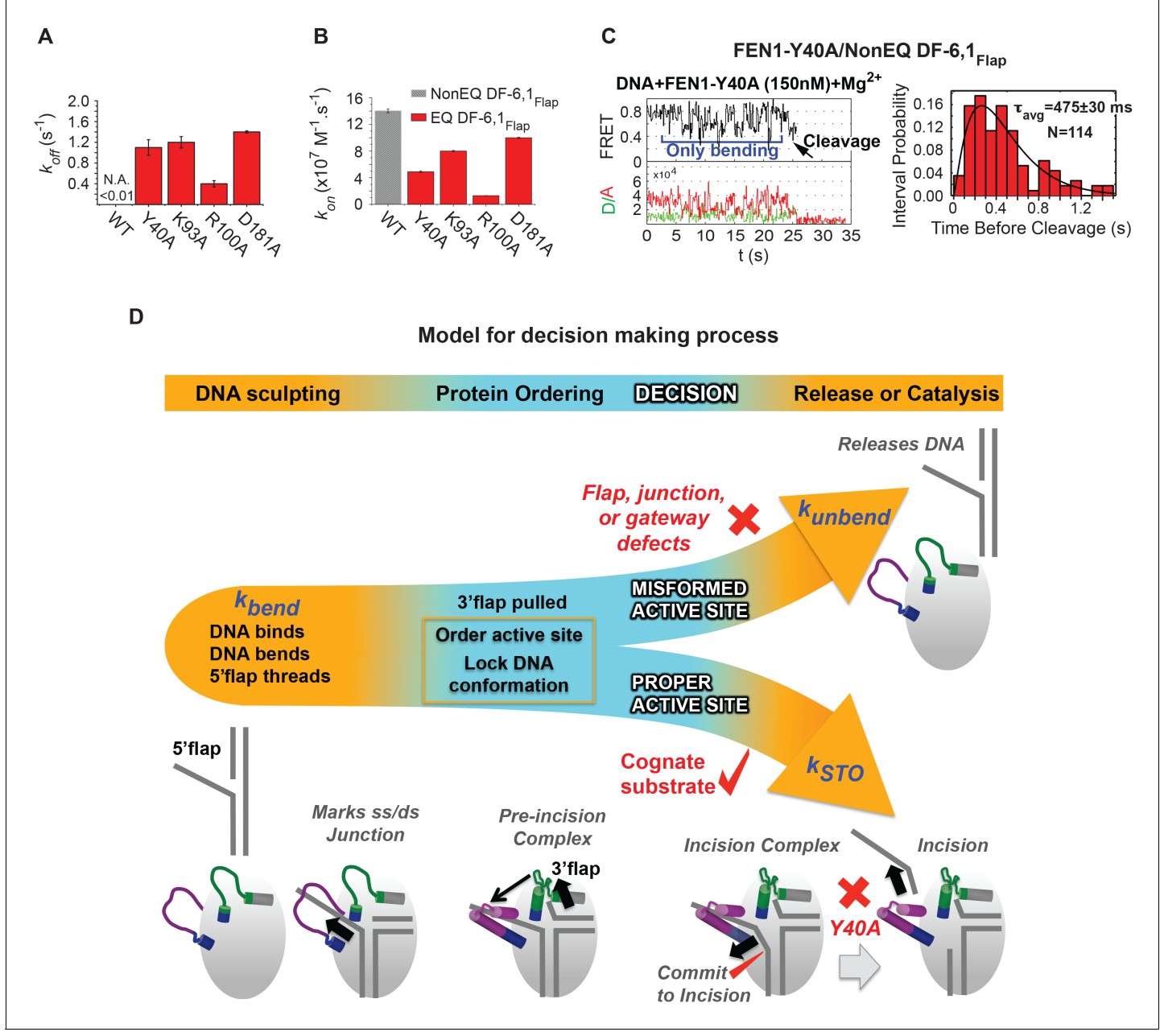

**Figure 6.** Role of active-site residues in active positioning of the 5'flap and the junction. (**A**) Bar chart comparing $k_{off-unbending}$ of WT-FEN1 and the FEN1 active-site mutants Y40A, K93A, R100A and D181A on NonEQ DF6,1$_{Flap}$. $k_{off-unbending}$ is calculated as described in 1G. (**B**) Bar chart comparing $k_{on-bending}$ of wild-type FEN1 on EQ DF6,1$_{Flap}$ and the FEN1 mutants Y40A, K93A, R100A and D181A on NonEQ DF6,1$_{Flap}$. $k_{on-bending}$ is calculated as described in **Figure 1G**. (**C**) A representative smFRET time trace showing the cleavage of NonEQ DF6,1$_{Flap}$ by FEN1-Y40A after multiple trials of DNA bending (left panel) and a histogram showing the distribution of dwell times of the bent state prior to cleavage (right panel). $\tau_{avg}$ from N = 4 is calculated as in **Figure 2C** at a temporal resolution of 50 ms. (**D**) Model for control of catalytic selectivity by the DNA mutual-induced fit mechanism in FEN1. DNA sculpting: FEN1 actively bend a variety of structures to verify the key features of its cognate DF substrates of fully paired ss/dsDNA nick junction, threaded 5'flap into the cap-helical gateway and 3'flap. Protein ordering: FEN1 actively pulls the 3'end of the nick junction to create a 3'flap and drive protein ordering, which in turn orders the active site and locks the DNA conformation. Decision: the active site and locked DNA conformer are always in catalytically competent form in cognate substrate, while they are primarily in catalytically incompetent form in noncognate substrates (no 5'flap threading, no 3'flap, mispair junctions) and FEN1 mutants (R47A, K93A and R100A). DNA release or catalysis: the DNA will shift or unpair to move the scissile phosphate into the active site for cleavage as probed by the flap/junction positioning-residue Y40, while in noncognate substrates FEN1 promotes DNA dissociation.

The following figure supplement is available for figure 6:

*Figure 6 continued on next page*

*Figure 6 continued*

**Figure supplement 1.** Sequences of DNA constructs used in this study.

figure supplement 1E). This highlights the extreme selectivity of FEN1 to local variation in the positioning of the junction and 5'flap for proper DNA lockdown. Unlike noncognate substrates or FEN1-R47A, the $k_{STO}$ in FEN1-Y40A increased by about three fold (*Figure 6C*); nonetheless, $k_{off\text{-}unbending}$ was still about two-fold slower than its $k_{STO}$ (*Figure 6A,C*). This indicates the presence of another step after proper DNA lockdown that involves active positioning of the scissile phosphate for incision.

## Discussion

Critical cellular processes such as DNA replication and repair are regulated by the molecular properties encoded in interacting macromolecules whereby distinct dynamic conformations correspond to different functional outcomes. The mechanisms for these dynamic changes that occur during macromolecular interactions are the subject of intense interest with two major proposed mechanisms, 'induced fit' and 'conformational selection'. Findings so far have suggested that many processes are regulated by conformational change before substrate binding by the substrate's selective binding to the active form of the enzyme, indicating that functional or conformational selection is in play (*Boehr et al., 2009*; *Vogt et al., 2014*). However the high precision required for DNA replication and repair has consistently raised issues of whether or not these might involve unusual mechanisms of chemistry or physics. In this context, how DNA-repair enzymes specifically recognize and remove damage in DNA is a decades-long debate. Does the damage destabilize the DNA duplex leading to disruption of the DNA structure (extrahelical base flipping or DNA bending) before its subsequent capture by the repair enzymes or do these enzymes actively sculpt the DNA as part of their recognition of the damage?

With the FEN1 single-molecule results, a picture emerges of induced conformational changes to both substrate and protein playing key roles in stabilizing a transition state that has been thoroughly vetted using multiple checks and is poised for catalysis with remarkable specificity (*Figure 6D*). In this process, FEN1 can differentiate between substrates whose incision is good for the cell (cognate) or toxic (noncognate) even if these substrates have small differences in their binding affinities. Active DNA bending does not create a significant energy barrier as evident by diffusion-limited on-rates in FEN1 and EXO1. We propose that different members of the 5'nuclease family share similar DNA-bending-induced disorder-to-order transitioning but differ in the mechanisms that couple this transitioning with active-site assembly. In FEN1, the coupling of protein transitions of the 3'flap-binding pocket and the 5'flap-binding helical gateway with DNA sculpting uncovers how dynamic protein segments are critical contributors to substrate binding and catalytic selection.

As part of the active DNA sculpting process, we observed single-molecule measurements consistent with a mutual induced-fit mechanism, with the protein bending the DNA and the bent DNA inducing a protein-conformational change (*Figure 6D*). Substrate distortion by ~90° and DNA-induced conformational changes in proteins are features that extend beyond 5'nucleases (*Gwon et al., 2014a*, *2014b*; *Yan et al., 2015*). More generally, the FEN1-type induced-fit mechanism may be central to detecting chemically subtle but biologically critical differences between correct and incorrect substrates for multiple DNA and RNA processes. Here, nuclease precision in replication comes from induced fit that regulates the compatibility of the distorted DNA conformer with active-site assembly and its off rate to allow cleavage of cognate but not noncognate substrates (*Figure 6D*).

The diffusion-limited bending of the cognate substrate by FEN1 and its cleavage from the first encounter represents a practically perfect precision reaction whose rate is limited by diffusion. The stochastic cleavage behavior of noncognate substrates after multiple cycles of DNA bending and dissociation of FEN1 has a fundamental bearing on how enzyme specificity is understood. In cases in which an enzyme encounters noncognate substrates and cleaves them after multiple trials, results on substrate specificity from classical biochemical techniques that monitor product formation become misleading. Furthermore, multiple attempts to cleave noncognate substrates are likely to be

insignificant inside the cell. These findings advance our insight into a previously unidentified mechanism in structure-specific nucleases for extreme specificity towards their cognate substrates inside the cell.

The 3'flap binding pocket is distant from the active site, raising the question why FEN1, along with some other structure-specific nucleases, utilizes long-range DNA-induced conformational coupling, in contrast with local coupling as observed in EXO1 (*Gwon et al., 2014a*, *2014b*; *Orans et al., 2011*; *Sakurai et al., 2005*; *Tsutakawa et al., 2011*; *Yan et al., 2015*). FEN1 cleaves 5'flaps containing RNA, DNA or mismatches of various lengths. We anticipate that long-range DNA-induced conformational coupling could provide a mechanism that enhances flexibility in nuclease substrates. The action of 3'flap-induced protein ordering as a key step that locks the FEN1 interaction with the junction could provide the advantage of limiting the sampling time between the disordered protein form and noncognate substrates that could otherwise lead to nonspecific cleavage. This active site control via a long-range induced-fit mechanism suggests why mutations distant from the FEN1 active site have a dramatic effect on genomic stability and disease states (*Sun et al., 2017*; *Zheng et al., 2007*). Thus, evolutionary selection against toxic and mutagenic DNA instability may have developed the unusual DNA-induced conformational coupling seen in FEN1 as a previously unrecognized part of repair and replication nuclease fidelity.

## Materials and methods

### Protein purification

Human FEN1 (amino acids: 2–380) was cloned into a pE-SumoPro expression vector (Lifesensors), which encodes an N-terminal His$_6$-Tag followed by SUMO protein. This clone was used for recombinant expression FEN1 in BL21(DE3) *E. coli* strain. FEN1 was purified by multiple chromatographic steps involving two sequential Ni-NTA columns interspersed by sumo protease cleavage. Purity was further increased by running FEN1 on heparin column and Hioload superdex-75 gel filtration column. FEN1 was dialyzed against buffer (50 mM Tris-HCl pH 7.5, 50% glycerol, 300 mM NaCl, 10 mM BME), flash frozen and stored at −80°C.

Human EXO1 catalytic domain (amino acids: 2–352; referred to as EXO1 in this study) was cloned into a pE-SumoPro expression vector (Lifesensors). EXO1 was expressed and purified using a similar protocol to that described for FEN1. The purified EXO1 was dialyzed against buffer (50 mM Tris-HCl pH 7.5, 50% glycerol, 300 mM NaCl, 10 mM BME), flash frozen and stored at −80°C.

### DNA substrate preparation

DNA oligos, modified and unmodified, were purchased from Integrated DNA Technologies (IDT). All sequences used to make the substrate structures are shown in *Figure 6—figure supplement 1*. To prepare the DNA substrates, we mixed equimolar concentrations of oligos in TE-100 and annealed by heating to 95°C followed by slow cooling to room temperature. Properly annealed substrates were purified on non-denaturing PAGE gel and extracted using the crush and soak method followed by ethanol precipitation.

### TIRF-based smFRET

Experiments were performed in a microfluidic flow chamber made by sandwiching a polyethylene spacer (100 μm thick polyethylene double-sided spacer SA-S-1L from Grace Biolabs) between a quartz slide and a glass coverslip with inlet and outlet tubing channels. The glass coverslip was functionalized and passivated by a combination of 1:100 molar ratio of biotinylated polyethylene glycol (Biotin-PEG-SVA MW 5,000) and polyethylene glycol (mPEG-SVA MW 5000) (Laysan Bio Inc.). The flow chamber was incubated with NeutrAvidin (0.2 mg/ml) just prior to the experiment for 2 min and then washed excessively with reaction buffer (50 mM HEPES pH 7.5, 1 mM DTT, 5% glycerol and 0.1 mg/mL BSA, 100 mM KCl and 10 mM CaCl$_2$); in the cleavage assays, CaCl$_2$ was replaced with 10 mM MgCl$_2$. This was followed by incubation with biotin-labeled DNA substrate (100 pM) until a sufficient surface coverage of fluorescent-labeled substrate was achieved. This was followed by washing with imaging buffer (described below) containing the appropriate divalent metal ion, CaCl$_2$ for bending or MgCl$_2$ for bending and cleavage.

To minimize the effect of photobleaching and photo-blinking, we used an oxygen scavenging solution as described earlier (*Aitken et al., 2008*) leading to the enzymatic removal of oxygen by a 6 mM proto-catechuic acid (PCA) (Sigma-Aldrich, Japan, P5630) and 60 nM protocatechuate-3,4-dioxygenase (PCD) system. Trolox (Sigma-Aldrich) was added at 2 mM concentration to reduce the photo-blinking by quenching the triplet state. The imaging buffer contained the reaction buffer and the aforementioned oxygen scavenging solution.

The experiments were performed on a custom-built TIRF-FRET setup as described earlier (*Sobhy et al., 2013*). For data analysis, the spatial mapping of the donor and acceptor emission channels was first calibrated using fluorescent beads that were imaged in TIRF mode. This generated a transformation matrix file, which was then used in the subsequent analysis of fluorescent molecules to map the donor and acceptor positions. The fluorescent molecules were registered as Gaussian point spread functions (PSF) around the brightest pixel in both channels and aligned with each other using the transformation matrix file. Donor and acceptor intensities were extracted by using software as described previously (*Holden et al., 2010*) and the apparent FRET efficiency was subsequently calculated. Any molecules with aberrant emission in brightness were excluded from further analysis. Histograms of FRET efficiencies were obtained from alternating the excitation of donor and acceptor 2c-ALEX as described previously (*Kapanidis et al., 2005*). The vbFRET package implemented in Matlab (*Bronson et al., 2009*) was used for dwell time analysis. The association ($\tau_{bending}$) and disassociation ($\tau_{unbending}$) dwell times were generated by idealizing and fitting the single molecule traces with two FRET states modeled by vbFRET (bent and unbent states). Histograms that were generated from dwell times in each state were fit with a single exponential decay function to generate $k_{bending}$ ($1/\tau_{unbending}$) and $k_{unbending}$ ($1/\tau_{bending}$), respectively.

## Confocal-based smFRET

The experiments were performed on a custom-built confocal epifluorescence microscope setup (*Cotlet et al., 2005*). Fluorophores were excited with a 532 nm line of a pulsed laser diode operating at 20 MHz (100 ps pulse width, LDH-P-FA-530L, PicoQuant) or a 50 mW 532 nm Cobolt Samba laser through a microscope objective. A water immersion objective (UPLSAPO60XW NA 1.2, Olympus) and an oil immersion objective (UPLSAPO100XO NA 1.4, Olympus) were used for solution-phase smFRET and smFRET on surface-immobilized molecules, respectively. A circularly polarized beam was obtained by inserting a Berek compensator (Mo. no. 5540, Newport) in the excitation beam path. The laser beam was made Gaussian and expanded to fill the back aperture of the objective lens before introducing it into the microscope using a spatial filter with a 30 μm pinhole. The laser beam was reflected off the surface of a longpass dichroic Di02 R532−25 × 36 (Semrock Inc.) into the objective. The excitation power at the sample plane was set to 400 Wcm$^{-2}$ or 255 Wcm$^{-2}$ for solution-phase smFRET or smFRET on surface-immobilized molecules respectively. In the detection path, emitted fluorescence passed through the dichroic and was focused onto a 100 μm pinhole by the tube lens of the microscope (IX71, Olympus) and recollimated using a lens. A longpass BLP01-532R-25 filter (Semrock Inc.) was used to remove scattered laser light, and then the beam was split into donor and acceptor channels using a dichroic FF635-Di01−25 × 36 (Semrock Inc.). The donor and acceptor paths were equipped with a band-pass FF01-580/60-25-D (Semrock Inc.) and a longpass LP02-664RU-25 filter (Semrock Inc.), respectively, before being focused onto single-photon avalanche diodes (τSPAD, PicoQuant). Fluorescence intensity trajectories were recorded by a time-correlated single-photon counting (TCSPC) module (HydraHarp 400, PicoQuant) in the time-tagged time-resolved (TTTR) mode, which allowed for recording the arrival time of each photon emitted by the fluorophores. The SymPhoTime software (PicoQuant) was used for the data acquisition as well as for controlling the excitation lasers and TCSPC module.

The solution-phase smFRET experiments were performed in a home-made flow-through chamber by sandwiching a paraffin film spacer (0.13 mm thick, Bemis Inc.) between two glass coverslips. The glass coverslips were functionalized and passivated with polyethylene glycol (mPEG-SCA, MW5000) (Laysan Bio Inc.) prior to the construction of the flow-through chamber. Samples were prepared by mixing imaging buffer with appropriate dilutions of the stock enzyme and stock DNA solutions. The solution was allowed to flow onto the flow cell by pipetting the solution into one side of the chamber while applying suction to the opposite end. Then, the flow cell was placed on the microscope. In all solution-phase smFRET experiments, the reaction buffer was used with the addition of 10 mM $CaCl_2$ and 2 mM Trolox. In the solution-phase smFRET experiments, the excitation laser was focused

approximately 40 µm above the surface of the bottom cover slip. The fluorescence intensity trajectories on the donor and acceptor channels were recorded for 15 min to obtain between 3500–8000 bursts from individual DNA molecules. SymPhoTime script was used to analyze the bursts and generate burst histograms. The intensity trajectories were first binned to 0.5 ms, and bursts above 35 total counts were considered for the analysis. The FRET efficiency was calculated by the integrated intensity of each burst in the donor and acceptor channels. OriginPro was used to fit the histograms of the FRET efficiency to Gaussian peaks.

The smFRET experiments on surface-immobilized molecules were performed using either the microfluidic flow chamber used in the TIRF-based FRET experiments or pre-made sticky-Slide VI$^0$.4 microfluidic chambers (ibidi GmbH), with cover slips identical to those used in the TIRF-based FRET experiments. The DNA substrates were immobilized on the glass cover slips according to the procedures described above. The smFRET experiments were performed in the presence of the oxygen scavenger and the triplet quencher used in the TIRF-based FRET experiments. The excitation laser was focused on the surface of the cover slip using back reflection. Fluorescence intensity trajectories of individual molecules were acquired by first scanning a 10 × 10 µm section of the coverslip using a scanning piezo stage. Then, individual molecules were manually chosen from the image and the trajectories were sequentially acquired, with the laser focus dwelling on each point for 10 s. The SymPhoTime software was used for the image acquisition and stage positioning.

Once fluorescence intensity trajectories were acquired, SymPhoTime was used to generate traces by binning the data to either 2, 5 or 10 ms and then exporting the donor and acceptor counts. A custom-written MATLAB script was used to generate traces from data exported from SymPhoTime and subsequently to select regions before photobleaching (*Harris, 2017*) a copy is archived at https://github.com/elifesciences-publications/ConfocalFret). Then, the FRET efficiency trace was calculated using the intensity trajectories of the donor and acceptor, and the histograms of the FRET efficiency were generated from the selected regions of the traces. Aberrant traces were excluded for further analysis. The selected regions were exported by the MATLAB script into files readable by HaMMy, a software used for analysis of single-molecule FRET trajectories using hidden Markov modeling (*McKinney et al., 2006*). The FRET trajectories were analyzed by a two-state model using HaMMy. Another custom-written MATLAB script was then used to collate the results from HaMMy and generate lists of dwell times for low FRET (that is, bent) and high FRET (that is, unbent) states. These lists were imported into OriginPro, histogrammed and fitted to a single exponential decay.

## Single-molecule cleavage assays

Cleavage experiments were performed by TIRF-based smFRET at a temporal resolution of either 50 or 100 ms. The surface-immobilized substrate was pre-incubated in the flow chamber with imaging buffer containing 10 mM MgCl$_2$. In the case of cleavage of NonEQ DF6,1$_{Flap}$ and EQ DF6,1$_{Flap}$ by wild-type FEN1, image acquisition started before FEN1 reached the microfluidic chamber. In all other cleavage experiments, imaging started after the protein had reached the microfluidic chamber but before it reached the focal volume. This delay in acquisition was to reduce particle loss due to acceptor photobleaching since the waiting time before cleavage markedly increased under these suboptimal conditions. Selective loss of the donor signal was confirmed by direct excitation of the acceptor at the end of the cleavage experiment.

Due to short lag time before cleavage, manual counting of frames in the bent state was used to calculate the cleavage dwell time for each trace. All particles in the field of view were grouped into the following five different categories as shown in *Figure 2—figure supplement 2A*. (a) Molecules with aberrant intensity that suffered from strong noise, photoblinking, step bleaching or deviation from the average intensity. These particles were excluded from further analysis. (b) Molecules that had acceptor photobleaching before the loss of the donor signal. These molecules do not influence dwell time analysis, which depends only on the donor signal. They were therefore excluded. (c) Molecules that went to the low FRET bent state followed by a single-step loss in the donor signal. This formed the bulk of the traces of cognate substrates. (d) Molecules that lost their donor signal without going into the low FRET bent state. This excluded minority population could result from donor photobleaching events and/or 5′flap cleavage that occurs at a faster rate than the acquisition time. (e) Molecules that stayed in the unbent high-FRET state within our imaging time. These molecules were also excluded from the cleavage analysis as they exhibit no FEN1 binding. The following criteria were used to perform dwell time analysis for the selected particles. (a) A minimum FRET change

of 0.2 between unbent and bent frames was applied as a filtering criterion before selecting traces for dwell time analysis. (b) Each selected trace was checked for anti-correlated behavior between the donor and acceptor upon change in the FRET efficiency. (c) With noncognate substrates, the dwell time was calculated by counting the number of frames spent in the lower FRET state before the donor signal was lost in the last bent step. MATLAB was used to calculate the mean of the cleavage dwell time by fitting with gamma distribution function and the error in the mean by bootstrap. In the viscosity measurement, a falling balls viscometer (Gilmont) was used to calculate the absolute viscosity. The density of the solution was measured from the mass of 1 ml of the same solution as used to calculate the viscosity.

## Time-resolved bulk FRET measurements

The measurements were performed on a QuantaMaster 800 spectrofluorometer (Photon Technology International Inc.) coupled with a supercontinuum fiber laser source. The fluorophore lifetime was determined by the time-correlated single-photon counting (TCSPC) method. The excitation was carried out at 535 nm and emission was collected at 568 nm using 5 nm bandwidths for the excitation and emission. A longpass filter with 550 nm cut-on was placed on the emission side to prevent scattered light. The instrument response function (IRF) was determined using a colloidal silica suspension. The decay time traces were acquired at 10,000 counts. The measurements were performed at room temperature using the same 5′ flap constructs and buffer composition as in the smFRET experiments. The determination of the lifetimes was done using IRF reconvolution and a multi-exponential decay function incorporated in the FluoFit software package (PicoQuant). The donor lifetime curve was fitted to two-exponential decay. The best fit was selected based on reduced chi-square and randomness of the residuals.

The FRET efficiency, $E_{FRET}$, refers to the conformational change resulting from the action of the enzyme on the substrate. $E_{FRET}$ is calculated from the measured lifetime of the donor in the donor-only and donor-acceptor substrates at the respective enzyme concentration using the following Equation:

$$E_{FRET} = 1 - \left( \frac{tau_{DA-Enzyme}}{tau_{D-Enzyme}} \right)$$

where $tau_{D-Enzyme}$ and $tau_{DA-Enzyme}$ are the amplitude-weighted average lifetimes of the donor excited-state in the donor-only and donor-acceptor substrates in the presence of the enzyme, respectively.

The dissociation bending constant ($K_{d-bending}$) was calculated by fitting the data to a standard quadratic equation for simple bimolecular association (S+F ⇌ SF) under equilibrium conditions:

$$E = E_0 + \left( E_f - E_0 \right) \left( \frac{\left( S + F + K_{d-bending} \right)^2 - \sqrt{\left( S + F + K_{d-bending} \right)^2 - 4SF}}{2S} \right)$$

where $E$ is the FRET value at any protein concentration, $E_0$ and $E_f$ are the initial and final FRET values, and S and F represent the substrate and FEN1 concentrations, respectively.

## Surface plasmon resonance (SPR) binding

SPR binding was performed on a Biacore T100 (GE Healthcare Inc.). Biotinylated DNA substrates were immobilized on S-series streptavidin sensor chips in HBS-EP buffer according to the manufacturer's recommendations. The response unit (RU) of the immobilized substrate is stated in the figure legends. FEN1 was dialyzed overnight at 4°C against the smFRET reaction buffer containing $CaCl_2$ (no oxygen-scavenging solution was added). Serial dilutions of FEN1 were made using the same reaction buffer. For each concentration, the run started with a surface-regeneration injection of reaction buffer +1 M NaCl at a flow rate of 100 µL/min for 120 s, followed by protein sample injection at a flow rate of 20 µL/s for 90 s for DF-6,1 or 20 µL/s for 120 s for the other substrates. The sensorgrams were corrected for bulk refractive index and residual nonspecific binding to the surface using a blank flow cell. The sensorgrams were processed using Biacore T100 Evaluation Software (GE Healthcare Inc.). The maximum RUs reached at each FEN1 concentration were fitted using the steady-state affinity mode to obtain the equilibrium dissociation constant ($K_{d-binding}$) for each DNA substrate.

## Steady-state bulk-cleavage assays

Reaction mixtures containing 0.5 nM Cy5-labeled NonEQ DF-6,1$_{dsDNA}$ (Cy5 was placed 15-nt away from the nick junction on the downstream dsDNA on the 5'flap primer) in 1X reaction buffer (50 mM HEPES-KOH pH 7.5, 100 mM KCl, 0.1 mg/ml BSA, 5%(v/v) glycerol, 10 mM MgCl$_2$ and 1 mM DTT) were pre-incubated at 37°C before the initiation of the cleavage reaction with the addition of varying concentrations of FEN1. Each reaction mixture was incubated further at 37°C and equal aliquots were removed and quenched by equal volumes of 2X denaturing buffer (90% deionized formamide, 100 mM EDTA) at the following time intervals (0, 0.17, 0.5, 1, 1.5, 2, 5, 10 mins). These samples were run on 20% denaturing PAGE gels, which were imaged using a Typhoon TRIO Variable Mode Imager (GE Healthcare, Life Sciences). The product formation was quantified using the *ImageJ* gel analysis tool. For each FEN1 concentration, the concentration of the product formed was plotted against time to estimate the initial rate ($v_0$, nM.min$^{-1}$) by taking the slope of the linear part. These $v_0$ values were plotted against the FEN1 concentration and $K_m$ was determined by nonlinear least-squares fitting using a Michaelis-Menten model.

## Molecular dynamics simulations

All simulations were performed with the AMBER 15.0 molecular dynamics package using the Parm14 force field with parmbsc0 nucleic acid modifications (*Case et al., 2015*; *Maier et al., 2015*). In total, four substrate DNA models (denoted NonEQ DF-6,1, SF-6,0, EQ DF-6,1 and nicked DNA) were generated. A dsDNA 47-mer with sequence d(5'-TGACCGTTGTTTGACGGTCGTGAGGAGGAAAG TTCCTCCTACGGCAG-3')•d'(5'-CTGCCGTAGGAGGAACTTTCCTCC**T(25)C(26)A(27)**CGACCG TCAAACAACGGTCA-3'), identical to the one used in the smFRET experiments, was first constructed in a canonical B-DNA conformation with the UCSF CHIMERA program (*Pettersen et al., 2004*). The model NonEQ DF-6,1 was built by adding 5' ssDNA flap (5'-TTTTTA-3') and a single-nucleotide 3' flap (G-3') at the junction of the two DNA duplexes (between bases C26 and T25). The SF-6,0 and EQ DF-6,1 models were constructed by adding 5' flap ssDNA (5'-TTTTA-3') at base C26 and 5' flap ssDNA (5'-TTTTAC-3') at base A27, respectively. Each system was solvated with TIP3P water (*Jorgensen and Jenson, 1998*) with a minimum distance of 15.0 Å from the DNA to the edge of the periodic simulation box. The systems were then neutralized by the addition of Na$^+$ counterions. Additionally, 100 mM NaCl concentration was introduced to mimic physiological conditions. First, the water and ions were subjected to 3000 steps of steepest descent and 1500 steps of conjugate gradient minimization while restraining all DNA atoms with a force constant of 2 kcal/mol Å$^2$. All restraints were then released. The particle mesh Ewald (PME) method (*Darden et al., 1993*) was used to treat the long-range electrostatic interactions. The cutoff for non-bonded interactions was set to 10 Å. All bonds involving hydrogen atoms were constrained using the SHAKE algorithm. We imposed a 1-fs simulation time step during equilibration. The temperature of the simulated systems was then gradually increased to 300 K over 50 ps in the NVT ensemble. Subsequently, equilibration dynamics was carried out in the NPT ensemble (p=1 atm and T = 300 K) for an additional 50 ns. Then, 800-ns production runs were carried out for each simulation system in the isothermal isobaric ensemble (p=1 atm and T = 300 K). We utilized hydrogen mass repartitioning (HMR) as a method to increase the simulation time step to 4 fs during the production runs (*Hopkins et al., 2015*). The substrate-bending angle for each system was defined and computed as described in *Figure 1—figure supplement 3A*. Data were analyzed with the CPPTRAJ code in AMBER15 (*Case et al., 2015*) and TCL scripts in VMD (*Humphrey et al., 1996*).

The effective free-energy profile (potential of mean force; PMF) for bending the NonEQ DF-6,1 DNA substrate was estimated using the adaptive biasing force (ABF) method (*Comer et al., 2015*; *Hénin et al., 2010*) with the COLVARS module of NAMD 2.11 (*Phillips et al., 2005*). ABF is a widely used enhanced sampling approach, which computes average forces along a predefined reaction coordinate (RC) and then applies an adaptive biasing potential to flatten the underlying free energy landscape. As a result, all points along the RC can be sampled efficiently. In our case, the RC was defined as the bending angle between the two-dsDNA fragments of the substrate. The exact definitions of the two vectors and the fragments are shown in *Figure 1—figure supplement 3A*.

First, we carried out a 20-ns targeted molecular dynamics (TMD) simulation (*Schlitter et al., 1994*), with a force constant of 100 kcal/mol Å$^2$ applied to all nucleic acid heavy atoms. This simulation transformed the DNA from a straight conformation (bending angle of ~180°) to the bent

conformation observed in the FEN1/DNA complex (bending angle of ~90°). The target configuration of the bent DNA was directly taken from the FEN1/DNA X-ray structure 3Q8L (*Tsutakawa et al., 2011*). In the ABF simulations, the reaction coordinate was segmented into nine discrete windows with a confining wall potential ($k$ = 50 kcal/mol) placed at the boundaries. Snapshots collected from the TMD trajectory were used to seed the ABF windows. Each window was further subdivided into small $0.2^0$ bins. Force averages were then accumulated into the bins and continuously updated in the course of the ABF simulation. Cancellation of the averaged forces through the gradual introduction of an adaptive bias led to enhanced sampling and overcoming of energy barriers along the RC. Since the instantaneous forces may fluctuate considerably, the application the adaptive bias was delayed until an adequate number of force samples was collected (2000 samples). PMF reconstruction was then accomplished by integration of the averaged forces from the bins.

## Acknowledgements

We thank Vlad-Stefan Raducanu for helpful discussions. The research reported here was supported by King Abdullah University of Science and Technology through core funding to SMH and a Competitive Research Award (CRG3) to SMH and JAT, as well as National Science Foundation CAREER award MCB-1149521 and National Institute of Health grant R01GM110387 to II JAT also acknowledges support of a Robert A Welch Chemistry Chair, the Cancer Prevention and Research Institute of Texas, and the University of Texas System Science and Technology Acquisition and Retention STARs program. Computational resources were provided in part by a National Science Foundation XSEDE allocation CHE110042 and through an allocation at NERSC supported by the US Department of Energy Office of Science contract DE-AC02-05CH11231. The authors declare no competing financial interests.

## Additional information

### Funding

| Funder | Grant reference number | Author |
| --- | --- | --- |
| King Abdullah University of Science and Technology | 2201 CRG3 | Fahad Rashid<br>Paul D Harris<br>Manal S Zaher<br>Mohamed A Sobhy<br>Luay I Joudeh<br>Hubert Piwonski<br>John A Tainer<br>Satoshi Habuchi<br>Samir M Hamdan |
| National Science Foundation | MCB-1149521 | Chunli Yan<br>Ivaylo Ivanov |
| NIH Clinical Center | R01GM110387 | Chunli Yan<br>Ivaylo Ivanov |
| U.S. Department of Energy | DE-AC02-05CH11231 | Samir M Hamdan |

The funders had no role in study design, data collection and interpretation, or the decision to submit the work for publication.

### Author contributions

FR, Conceptualization, Formal analysis, Supervision, Validation, Investigation, Methodology, Writing—original draft, Project administration, Writing—review and editing, Established the single-molecule TIRF experiments, Designed and performed the single-molecule TIRF experiments and purified proteins, Designed the confocal FRET experiments, Designed the study, Supervised the study and wrote the manuscript; PDH, Formal analysis, Investigation, Methodology, Designed and performed the confocal FRET experiments; MSZ, Formal analysis, Investigation, Methodology, Performed the EXO1 experiments, bulk cleavage assays and supported Fahad Rashid in optimizing and analyzing the single-molecule TIRF experiments and protein purification; MAS, Investigation, Methodology, Established the single-molecule TIRF experiments, Performed the time-resolved bulk FRET

experiments; LIJ, Investigation, Methodology, Performed the SPR experiments; CY, Investigation, Methodology, Performed MD simulations; HP, Investigation, Methodology, Designed the confocal FRET experiments; SET, Funding acquisition, Investigation, Writing—original draft, Designed the study; II, Funding acquisition, Investigation, Methodology, Writing—original draft, Performed MD simulations; JAT, Funding acquisition, Investigation, Writing—original draft, Writing—review and editing, Designed the study; SH, Supervision, Funding acquisition, Investigation, Writing—original draft, Writing—review and editing, Designed the confocal FRET experiments, Designed the study; SMH, Conceptualization, Formal analysis, Supervision, Funding acquisition, Validation, Investigation, Methodology, Writing—original draft, Project administration, Writing—review and editing, Designed the study. Supervised the study and wrote the manuscript

### Author ORCIDs
Hubert Piwonski, http://orcid.org/0000-0001-8666-3945
Samir M Hamdan, http://orcid.org/0000-0001-5192-1852

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
