## [Decision Letter]

Thank you for submitting your article "Single-Molecule Complexes Unveil Utmost Nuclease Accuracy in DNA Replication and Repair" for consideration by *eLife*. Your article has been reviewed by two peer reviewers, and the evaluation has been overseen by a Reviewing Editor and Jessica Tyler as the Senior Editor. The reviewers have opted to remain anonymous.

The reviewers have discussed the reviews with one another and the Reviewing Editor has drafted this decision to help you prepare a revised submission.

Summary:

The present paper uses single-molecule FRET methods, complemented by other ensemble-averaging biophysical approaches, to understand how the flap endonuclease FEN1 interacts with its substrate. In particular, the authors focus on the classical question of induced fit or conformational selection as the two main models that describe recognition of a substrate and productive interaction with it. A series of elegant single-molecule experiments are described that provide great detail of the kinetic steps underlying flap cleavage with a number of different cognate and noncognate substrates. The main conclusion is that both protein and DNA undergo conformational changes upon binding to ensure great selectivity and specificity in enzymatic activity. The work is comprehensive and detailed, and other complementary tools such as MD simulations, ensemble time resolved FRET, biochemical cleavage assays and surface plasmon resonance are employed to support their findings and interpretations. The results are intriguing and the findings are very relevant to the field, specifically for the area of DNA replication/ repair and for the broader area of enzymology. However, a number of important arguments and conclusions in the manuscript are not backed up by the data shown. Discrimination between the different models and scenarios requires high accuracy in the quantitative determination of FRET levels, transition rates etc. While the numbers in the manuscript seem to make sense, the underlying raw data, distributions, and analyses to arrive at these numbers are either not shown or not adequately explained. Before a decision can be made regarding publication, a revised manuscript should submitted that addresses the following comments:

Essential revisions:

– A major disagreement exists between the present manuscript and one of the authors' previous reports (Sobhy et al., Cell Reports 2013). Here, human FEN1 binds with about 5 nM Kd to various DNA substrates whereas in the previous work, the Kd was about 50 times higher. In addition, k-bending/binding is diffusion limited here, whereas in the previous work it is orders of magnitude lower. This discrepancy needs to be resolved.

– Along these lines, the authors claim in the previous work that, "we find a multistep mechanism that verifies all substrate features before inducing the intermediary-DNA bending step that is believed to unify 5' nuclease mechanisms. This is achieved by coordinating threading of the 5' flap of a nick junction into the conserved capped-helical gateway, overseeing the active site, and bending by binding at the base of the junction." These statements, if they are true, certainly steal some thunder from the current manuscript; however, in the absence of discussion of the previous results here, it is unclear whether or not the authors chose to disregard their previously published study because it was incorrectly performed. This issue needs resolution. In addition, the authors should cite a related single molecule FRET study of FEN1 by the Penedo lab published in Nucleic Acids Research 2014.

– Many of the critical arguments in the manuscript are based on the authors' ability to very precisely construct the distributions of times elapsed between bending and cutting. From the description in the manuscript it is not clear how this experimentally is exactly done. The authors need to clarify how they determine the exact moment of bending, and whether they exclude certain molecules based on the properties in their FRET traces.

– It is stated that the fraction of binding/bending events that result in cleavage is 100%. Please provide supporting data/statistics are provided to bolster this claim.

– In Figure 2 and Figure 5, the authors analyze the delay time between FRET decrease (bending) and disappearance of donor signal (flap cutting). How do they know the disappearance of donor signal is not caused by donor bleaching? The authors should provide experimental evidence to exclude this scenario.

– For non-optimal substrates, cleavage and product release is still observed, but only after many rounds of nonproductive binding/bending events. The authors treat the final binding event differently from the earlier binding events but why? A strong possibility is that each binding event can lead to cleavage but with a lower rate than in the case of optimal substrates. How is this possibility excluded? If it can be shown that the dwell time distribution of the final binding event is quantitatively different from the dwell time distributions of the earlier binding events, perhaps the current interpretation can be favored. Otherwise, the interpretation and with the last paragraph in subsection “FEN1 avoids off-target DNA cleavage in the DNA lockdown step” is suspect.

– In Figure 2, the authors show a rise-and-decay distribution of the times elapsed between bending and cutting. It is argued that this distribution is caused by the presence of multiple rate-limiting steps, including a disorder-order transition and cleavage chemistry. An important property of rise-and-decay distributions is that they only arise if the underlying steps are roughly equally rate limiting. In a subsequent, elegant experiment, the authors slow down the disorder-order transitions by a factor of 3-4 (Figure 2). This result provides a strong prediction; namely, that the rise-and-decay distribution should now collapse into a distribution that is entirely determined by the disorder-order transition as the rate-limiting step. However, when looking at Figure 2—figure supplement 1, a rise-and-decay distribution is seen with an even larger number of rate-limiting steps. This disconnect needs to be resolved.

– The argument for mutual induced fit is very speculative. Protein conformational changes are not measured directly and any conformational change inferred does not seem well supported by the data. It is strongly recommend that these claims be removed.

– The conclusions drawn based on the data in Figure 3 rely on the authors' ability to reliably distinguish fairly similar FRET levels. It is described how the confocal FRET data was analyzed to do so, but the data aren't actually shown. Given the high level of confidence needed to believe the accuracy of the FRET values, it is critical that the authors to show these data, the observed variation within the sets of three experimental replicates, and the fitting of the data.

[Editors' note: further revisions were requested prior to acceptance, as described below.]

Thank you for resubmitting your work entitled "Single-molecule FRET unveils induced-fit mechanism for substrate selectivity in flap endonuclease 1" for further consideration at *eLife*. Your revised article has been favorably evaluated by Jessica Tyler (Senior editor), a Reviewing editor, and two reviewers.

The manuscript has been improved but some of the comments seem to have been misunderstood and as a result, their revision in response did not adequately answered the questions. These few remaining issues (outlined below) need to be addressed before acceptance:

1) "For non-optimal substrates, cleavage and product release is still observed but after many rounds of nonproductive binding/bending events. The authors treat the final binding event differently from the earlier binding events but why? A strong possibility is that each binding event can lead to cleavage but with a lower rate than in the case of optical substrates. How do they exclude this possibility? If they can show that the dwell time distribution of the final binding event is quantitatively different from the dwell time distributions of the earlier binding events, perhaps their current interpretation can be favored."

In response to this comment, the authors compared the lifetimes of the bound state in the presence of calcium with the lifetime of the final bound state before cleavage. However, such a comparison does not address the issue at hand, because one cannot rule out the possibility that having calcium instead of magnesium may change the kinetics on non-optimal substrates. Because the non-optimal substrates show multiple binding events (1, 2,.…, n-1) before the nth binding event results in cleavage, they should build the histogram of dwell times of prior binding events and compare it to the dwell time histogram of the final event. Only if the two are substantially different, would this conclusion be supported.

2) "– In Figure 2, the authors show a rise-and-decay distribution of the times elapsed between bending and cutting. It is argued that this distribution is caused by the presence of multiple rate-limiting steps, including a disorder-order transition and cleavage chemistry. An important property of rise-and-decay distributions is that they only arise if the underlying steps are roughly equally rate limiting. In a subsequent, elegant experiment, the authors slow down the disorder-order transitions by a factor of 3-4 (Figure 2). This result provides a strong prediction; namely, that the rise-and-decay distribution should now collapse into a distribution that is entirely determined by the disorder-order transition as the rate-limiting step. However, when looking at Figure 2—figure supplement 1, a rise-and-decay distribution is seen with an even larger number of rate-limiting steps. This disconnect needs to be resolved."

Again, the authors seem to have misunderstood the comment. At least one of the statements in the original manuscript is wrong, but which one has not been identified.

3) "– The data analysis seems to assume that productive release occurs instantaneously because the dwell time between initial bending and fluorescence disappearance is interpreted as a reaction time. Please provide justification for this supposition."

This is essentially a non-answer because the disappearance of ssDNA flap is defined as cleavage ("the ssDNA flap has no interaction with FEN1, whose disappearance is defined as cleavage in our assay"). Of course, with such a definition, product release is simultaneous with cleavage. However, cleavage should really relate to the chemical reaction of backbone scission; there is no a priori reason to believe that the cleavage product should be released instantaneously. This issue still needs to be addressed.

4) Regarding the observation that upon addition of viscogen, the dwell-time distribution remains a rise and decay (which implies that multiple rate-limiting steps continue to exist), it is stated that the presence of visogen itself would introduce a more complicated free-energy landscape that would result in additional rate-limiting steps. While this is possible in principle, it is not clear that the width of the dwell-time distribution necessarily is caused by diffusion. This statement should be clarified or omitted.

---

## [Author Response]

*Essential revisions:*

*– A major disagreement exists between the present manuscript and one of the authors' previous reports (Sobhy et al., Cell Reports 2013). Here, human FEN1 binds with about 5 nM Kd to various DNA substrates whereas in the previous work, the Kd was about 50 times higher. In addition, k-bending/binding is diffusion limited here, whereas in the previous work it is orders of magnitude lower. This discrepancy needs to be resolved.*

Good point. We are unclear as to why K_d-bending_ was high in our previous study (Sobhy et al., Cell Reports 2013), but we suspect that it might have resulted from a faulty preparation of the oxygen scavenging system. Here we also optimized the purification of the native FEN1 by using cleavable SUMO-fusion tag. Both slower association and faster dissociation rates contributed to the higher K_d-bending_ as pointed out by the reviewers. Nonetheless, the FRET values of the unbent and bent states in the flap- labeling scheme, the only labeling scheme used in Sobhy et al., and the relative comparison of bending cognate and noncognate substrates remain valid findings as confirmed by the current study under optimal K_d-bending_. In the revised manuscript we added the following paragraph to address this point.

“It is unclear what caused the much higher K_d-bending_ reported in our earlier work (Sobhy et al., 2013), but we suggest that both slower association and faster dissociation rates were influences. Nonetheless, the FRET states of NonEQ DF-6,1Flap alone and when bent by FEN1 and the relative comparison of bending the cognate with the noncognate substrates are similar under low and high K_d-bending_ conditions as shown below.”

*– Along these lines, the authors claim in the previous work that, "we find a multistep mechanism that verifies all substrate features before inducing the intermediary-DNA bending step that is believed to unify 5' nuclease mechanisms. This is achieved by coordinating threading of the 5' flap of a nick junction into the conserved capped-helical gateway, overseeing the active site, and bending by binding at the base of the junction." These statements, if they are true, certainly steal some thunder from the current manuscript; however, in the absence of discussion of the previous results here, it is unclear whether or not the authors chose to disregard their previously published study because it was incorrectly performed. This issue needs resolution. In addition, the authors should cite a related single molecule FRET study of FEN1 by the Penedo lab published in Nucleic Acids Research 2014.*

Thank you for noting these points. We apologize for the too brief explanation linking the current findings with those from Sobhy’s et al. This is now addressed in what we hope is suitable detail in the revised manuscript.

Sobhy et al. used the flap-labeling scheme to address the relationship between 5’flap threading into the cap-helical gateway and DNA bending. This labeling scheme showed that blocking 5’flap threading by introducing biotin/streptavidin moiety on the 5’flap resulted in minor change in FRET compare to that when threading was allowed. Indeed this experiment is reproducible under optimal K_d-bending_ (data not shown). Sobhy et al. therefore proposed that 5’flap threading is strictly required for inducing DNA bending and consequently that DNA bending occurs at the conclusion of substrate recognition; the paragraph that is highlighted by the reviewers. In the current study, we relied on the internal-labeling scheme that directly reports on the geometry of the duplex DNA. This labeling scheme led to a different result, showing significant distortion of the duplex DNA when the 5’flap threading process was blocked. However, this distortion was still less than that observed when 5’flap threading was allowed. We also found that altering residues in the gateway decreased the on rate of DNA bending. Although these new results also suggest that 5’flap threading is required for proper distortion of the DNA, as originally proposed by Sobhy et al., the significant DNA bending reported in the current study made it difficult to conclusively rule out the possibilities that FEN1 might be threading the 5’flap after it significantly bends the DNA or while it bends the DNA. These results advance our knowledge on the relationship between 5’flap threading and DNA bending but they also highlight the complexity in understanding the exact timing mechanism of 5’flap threading relative to DNA bending. In the current study, we also performed kinetic analysis of on- and off-rates on non-cognate substrates, combined with high temporal resolution FRET measurements, and showed that FEN1 was able to bend both cognate and noncognate substrates but displayed remarkable selectivity to stabilize the bent conformer for cognate substrate. These results together with the significant distortion of the DNA upon blocking 5’flap threading are more consistent with a model that FEN1 bends the DNA and uses it to interrogate the features of the substrate. In the revised manuscript, we added discussion linking the findings on 5’flap recognition from the flap-labeling scheme in Sobhy et al. with the internal-labeling scheme in the current study. We also added the following conclusion paragraph that comments on the proposed models in both studies.

“Collectively these results demonstrate that FEN1 bends both cognate and noncognate substrates and that K_d-bending_ is higher for noncognate substrates. This is consistent with our previous findings under high K_d-bending_ conditions (Sobhy et al., 2013). They further showed that FEN1 stabilizes the cognate substrate through remarkable selectivity for its key features of a fully paired nick junction, a 3’flap and a 5’flap while promoting the dissociation of noncognate substrates. Our observation of FEN1’s ability to significantly bend the DNA in the blocked-threaded complex challenges our previous conclusion that 5’flap threading is strictly required to induce DNA bending (Sobhy et al., 2013). Our new results are consistent with a model in which FEN1 actively bends DNA to interact with the ss/ds-DNA junctions and subsequently verifies these interactions by the 3’flap-induced protein ordering.”

The reviewers appropriately requested that we reference the work of Cragg et al. (NAR 2014). This study proposed that double flap substrate alternates between bent and unbent states. We believe that relying on the flap-labeling scheme is the reason why Cragg et al. proposed that double flap substrate is a dynamic structure. We found that this dynamic nature is a result of local variation in the geometry of the 5’flap rather than altering the conformation of the duplex DNA when the 5’flap length exceeds 6 nt.

Reproduction of Cragg et al. experiments and their discussion were provided in Figure 1—figure supplement 2 in the original manuscript. In the revised manuscript, we added the following statement in the main text to address this point.

“The flap-labeling scheme was sensitive to variation in the Mg^2+^ ion concentration when the 5’flap length exceeded 6 nt (Figure 1—figure supplement 2). This could explain why previous work suggested that a double-flap substrate is dynamic structure (Craggs et al., 2014).”

*– Many of the critical arguments in the manuscript are based on the authors' ability to very precisely construct the distributions of times elapsed between bending and cutting. From the description in the manuscript it is not clear how this experimentally is exactly done. The authors need to clarify how they determine the exact moment of bending, and whether they exclude certain molecules based on the properties in their FRET traces.*

We appreciate the reviewer’s expert insights on these points. In the revised manuscript, we provide a more detailed description on the assignment of cleavage events and their analysis in the Methods section. We also provide an extra detailed analysis of all particles in the entire field of view for both cognate and non-cognate substrates in a new supplementary Figure 2—figure supplement 2. These analyses now demonstrate quantitatively our ability to assign the 5’flap cleavage event.

*– It is stated that the fraction of binding/bending events that result in cleavage is 100%. Please provide supporting data/statistics are provided to bolster this claim.*

We didn’t intend to make that claim, and we regret this misunderstanding. What we meant is that in the case of cognate substrate, all the effective particles judged to have been cleaved based on the criteria described in detail in the Methods section in the revised manuscript and shown in the new supplementary Figure 2—figure supplement 2 are cleaved on first bending step. We trust that the new supplementary Figure 2—figure supplement 2 and the detailed description of data analysis in the Methods section will remove this confusion on our part in the revised manuscript.

*– In Figure 2 and Figure 5, the authors analyze the delay time between FRET decrease (bending) and disappearance of donor signal (flap cutting). How do they know the disappearance of donor signal is not caused by donor bleaching? The authors should provide experimental evidence to exclude this scenario.*

Thank you for making this point. In the revised manuscript, we performed detailed analysis of the control experiments that compare Cy3 loss under condition where FEN1 bends the cognate DNA without cleaving it (Ca^2+^ ions) or bend it and cleave it (Mg^2+^ ions) to show that Cy3 signal loss is due to cleavage; new supplementary Figure 2—figure supplement 2. The Cy3 signal loss under Ca^2+^ was ~20% while the signal loss under Mg^2+^ was ~80%. Similar results were found for the non-cognate substrate as shown in new supplementary Figure 2—figure supplement 2.

We further analyzed the time dependent signal loss of Cy3 in cognate substrate in presence of Ca^2+^ or Mg^2+^ as shown in new supplementary Figure 2—figure supplement 2. In presence of Ca^2+^, the number of Cy3 lost was slow and saturated around 20%.

While in case of Mg^2+^, there is an abrupt change in the rate of signal loss at around 6 s that coincides with the arrival of FEN1 to the flow cell.

Taken together these analysis show that Cy3 signal loss is mostly observed due to cleavage and not photo-bleaching.

*– For non-optimal substrates, cleavage and product release is still observed, but only after many rounds of nonproductive binding/bending events. The authors treat the final binding event differently from the earlier binding events but why? A strong possibility is that each binding event can lead to cleavage but with a lower rate than in the case of optimal substrates. How is this possibility excluded? If it can be shown that the dwell time distribution of the final binding event is quantitatively different from the dwell time distributions of the earlier binding events, perhaps the current interpretation can be favored. Otherwise, the interpretation and with the last paragraph in subsection “FEN1 avoids off-target DNA cleavage in the DNA lockdown step” is suspect.*

The lifetimes of the bent conformer in non-cognate substrates were determined in the presence of Ca^2+^ and are shown in Figure 4. Indeed, these lifetimes were significantly different than that observed for the last bending step before cleavage in the presence of Mg^2+^, either longer or shorter. This difference served as the base for our interpretation of how FEN1 avoids the cleavage of non-cognate substrates.

Briefly, the relationship between the lifetime of the bent conformer in presence of Ca^2+^ and the dwell time of the last bent state before cleavage in presence of Mg2 can be categorized into two behaviors. In one group, the lifetime was rate limiting and therefore not supporting the time required for catalysis. In the second it was 3-4 folds longer than the time required for catalysis. If every abortive bending event represents a step where catalysis proceeds at slower rates, then the anticipation would be that the lag time in the last DNA bending event that led to cleavage would be longer in both categories than cognate substrate, particularly in the second category where the life time of the bent conformer is significantly longer that required for catalysis. What we observed is that the lag times before cleavage were different than those shown in Figure 4, and was quantitatively similar in both cognate and non-cognate substrates. A scenario where catalysis proceeds at slower rate is shown in the case of the active site mutant FEN1-Y40A where the lag time before cleavage was 3 times longer than that in cognate and non-cognate substrates.

*– In Figure 2, the authors show a rise-and-decay distribution of the times elapsed between bending and cutting. It is argued that this distribution is caused by the presence of multiple rate-limiting steps, including a disorder-order transition and cleavage chemistry. An important property of rise-and-decay distributions is that they only arise if the underlying steps are roughly equally rate limiting. In a subsequent, elegant experiment, the authors slow down the disorder-order transitions by a factor of 3-4 (Figure 2). This result provides a strong prediction; namely, that the rise-and-decay distribution should now collapse into a distribution that is entirely determined by the disorder-order transition as the rate-limiting step. However, when looking at Figure 2—figure supplement 1, a rise-and-decay distribution is seen with an even larger number of rate-limiting steps. This disconnect needs to be resolved.*

Here we clarify our logic. If we assume that the distribution of the lag time should reflect the rate-limiting 3’flap-induced conformational changes step, it is still possible that the underlying distribution of conformational changes upon addition of viscogen may become a more complex distribution in itself. This can be understood from a simple diffusion model where the rate of folding is inversely proportional to the solution viscosity. Assuming that without viscogen, the conformational change was distributed exponentially, it can be argued that upon addition of viscogen, the conformational changes would assume multi-exponential behavior due to different degree of friction provided to different residues in the protein by the viscogen and thus showing apparent multi-exponential steps in the dwell times of cleavage. In fact, if we plot the standard deviation of the dwell time vs viscosity, we found a near linear relationship (a new supplementary Figure 2—figure supplement 1), as would be predicted by a simple diffusion model. In simple terms this could imply that when adding viscogen, the energy bounds of folding get larger causing the distribution of individual rates of protein folding to show a greater standard distribution among themselves. In the revised manuscript, we discussed the rise and decay distribution of dwell times in the presence of viscogen as follow:

“The shape of the histograms in the presence of viscogen however remains rise and decay (Figure 2—figure supplement 1), suggesting that the 3’flap-induced protein ordering is likely to involve multistep processes that are being slowed down by the presence of viscogen. In fact, the standard deviation of kSTO was almost linearly proportional to the viscosity as would be predicted by a simple diffusion where different protein regions experience different friction upon increasing viscogen concentration (Figure 2—figure supplement 1).”

*– The argument for mutual induced fit is very speculative. Protein conformational changes are not measured directly and any conformational change inferred does not seem well supported by the data. It is strongly recommend that these claims be removed.*

We appreciate this thoughtful recommendation. In the revised manuscript, we have revisited our arguments about mutual induced fit and used instead induced fit only. We agree with the reviewer that we are not sensitive to protein conformational changes. We are sensitive however to the pulling of 1 nt 3’ flap, which from previous structural studies is established to induce the conformational changes in the protein. In the revised manuscript, we also clarified in a suggestive way why we think that the induce- fit relationship between FEN1 and DNA is of a mutual nature by relying on the argument that 3’flap is being actively pulled after DNA bending.

*– The conclusions drawn based on the data in Figure 3 rely on the authors' ability to reliably distinguish fairly similar FRET levels. It is described how the confocal FRET data was analyzed to do so, but the data aren't actually shown. Given the high level of confidence needed to believe the accuracy of the FRET values, it is critical that the authors to show these data, the observed variation within the sets of three experimental replicates, and the fitting of the data.*

This is a key point that we are pleased to improve. We have added a new supplementary figure to show all the histograms pertaining to the figure in a new supplementary Figure 3—figure supplement 1. In fact we repeated all the experiments presented in Figure 3 in order to add one more control experiment showing no effect of FEN1 on dsDNA. This resulted in minor changes in all of the original FRET readings due the slight differences in the laser alignment conditions of the confocal microscope. As can be seen in the new Figure 3, the readings are extremely consistent among themselves and with those in the originally submitted Figure 3 and have very low standard deviation.

[Editors' note: further revisions were requested prior to acceptance, as described below.]

*The manuscript has been improved but some of the comments seem to have been misunderstood and as a result, their revision in response did not adequately answered the questions. These few remaining issues (outlined below) need to be addressed before acceptance:*

*1) "For non-optimal substrates, cleavage and product release is still observed but after many rounds of nonproductive binding/bending events. The authors treat the final binding event differently from the earlier binding events but why? A strong possibility is that each binding event can lead to cleavage but with a lower rate than in the case of optical substrates. How do they exclude this possibility? If they can show that the dwell time distribution of the final binding event is quantitatively different from the dwell time distributions of the earlier binding events, perhaps their current interpretation can be favored."*

*In response to this comment, the authors compared the lifetimes of the bound state in the presence of calcium with the lifetime of the final bound state before cleavage. However, such a comparison does not address the issue at hand, because one cannot rule out the possibility that having calcium instead of magnesium may change the kinetics on non-optimal substrates. Because the non-optimal substrates show multiple binding events (1, 2,.…, n-1) before the nth binding event results in cleavage, they should build the histogram of dwell times of prior binding events and compare it to the dwell time histogram of the final event. Only if the two are substantially different, would this conclusion be supported.*

We appreciate the insight from our reviewers. We compared dwell time of missed events in presence of Mg^2+^ in all noncognate substrates as well as the FEN1 mutants Y40A and R47A and found them to be similar to Ca^2+^ data. These new data are provided in a new panel in Figure 5—figure supplement 1 (panel F). The original conclusion therefore stands since the dwell-times are similar in both Mg^2+^ and Ca^2+^.

*2) "– In Figure 2, the authors show a rise-and-decay distribution of the times elapsed between bending and cutting. It is argued that this distribution is caused by the presence of multiple rate-limiting steps, including a disorder-order transition and cleavage chemistry. An important property of rise-and-decay distributions is that they only arise if the underlying steps are roughly equally rate limiting. In a subsequent, elegant experiment, the authors slow down the disorder-order transitions by a factor of 3-4 (Figure 2). This result provides a strong prediction; namely, that the rise-and-decay distribution should now collapse into a distribution that is entirely determined by the disorder-order transition as the rate-limiting step. However, when looking at Figure 2—figure supplement 1, a rise-and-decay distribution is seen with an even larger number of rate-limiting steps. This disconnect needs to be resolved."*

*Again, the authors seem to have misunderstood the comment. At least one of the statements in the original manuscript is wrong, but which one has not been identified.*

We apologize for the misunderstanding. We agree with the reviewers that rise and decay profile of dwell times suggests multiple steps with similar rate constants occurring during the dwell time. As the reviewers pointed out, when the rate constant of one of the steps (e.g. conformational change) decreases while others remain constant, that step acts as the ratelimiting step and the histogram should exhibit a single-exponential decaying profile. When this step slows down, the mean dwell time should become longer and this indeed what we observed experimentally in Figure 2. The standard deviations in the absence or presence of different glycerol concentrations were in the range of 60-70% of their mean, suggesting that the shape of the histograms is similar and that the number of rate limiting steps did not increase in presence of viscogen. This similarity in the rise and decay fits can be visually seen should we bin our data according to some optimal binning rule like Scott’s rule. However, we have decided to present them at the experimental temporal resolution since it reflects the closest representation of the raw data.

The reviewers are therefore correct that there are still multiple rate-limiting steps contributing to the cleavage reaction. One possibility is that there are multiple distinct steps of conformational changes in this reaction that will slow down when the viscosity of the solution increases. This includes the 3’flap-induced protein ordering that in itself could be multiple steps and potentially subsequent conformational changes that are required to unpair the DNA and shift it into the active site. If this is the case, all the steps will slow down when the viscosity of the solution increases, and therefore, overall reaction rate will decrease while the rise-and-decay shape of the histogram will remain the same. We therefore believe the statement in the original manuscript that 3’flap-induced protein ordering is the probable rate-limiting step is incorrect. In the revised manuscript we corrected this statement and also commented on the shape of the histograms as pointed out by the reviewers. The paragraph now reads as follows: “The shape of the histograms in the presence of viscogen however remains a rise and decay (Figure 2—figure supplement 1), in contrast to the prediction of collapsing into a single exponential decay should protein-ordering act in a single rate-limiting step. This suggests that the 3’flap-induced protein ordering likely involves multistep processes that are being slowed by the presence of viscogen and/or these multisteps control different rate-limiting steps during catalysis such as DNA unpairing and/or DNA shifting into the active site.”

*3) "– The data analysis seems to assume that productive release occurs instantaneously because the dwell time between initial bending and fluorescence disappearance is interpreted as a reaction time. Please provide justification for this supposition."*

*This is essentially a non-answer because the disappearance of ssDNA flap is defined as cleavage ("the ssDNA flap has no interaction with FEN1, whose disappearance is defined as cleavage in our assay"). Of course, with such a definition, product release is simultaneous with cleavage. However, cleavage should really relate to the chemical reaction of backbone scission; there is no a priori reason to believe that the cleavage product should be released instantaneously. This issue still needs to be addressed.*

Our definition of cleavage, and the conventional one used in literature, is from the moment FEN1 binds the DNA, the complex undergoes all the necessary conformational changes, the incision takes place and the 5’flap departs the complex. We provided evidences that are supported by previous literature that 5’flap release is not a limiting step, but certainly that doesn’t mean it is not contributing to the dwell time that we refer to as a single turnover. However, the strongest evidence for our proposition that majority of the dwell time before cleavage corresponds to catalysis rather than 5’flap release is that our single turnover is similar to those from bulk cleavage assays that are not sensitive to 5’flap release in case of cognate substrate. To remove any potential confusion in the definition of single turnover, we added the following statement in our revised manuscript: “However, since 5’flap release would still contribute to the dwell time before cleavage, our single turnover should be treated as an apparent value.”

4) Regarding the observation that upon addition of viscogen, the dwell-time distribution remains a rise and decay (which implies that multiple rate-limiting steps continue to exist), it is stated that the presence of visogen itself would introduce a more complicated free-energy landscape that would result in additional rate-limiting steps. While this is possible in principle, it is not clear that the width of the dwell-time distribution necessarily is caused by diffusion. This statement should be clarified or omitted.

We have omitted this statement from the revised manuscript as suggested and also the associated figure panel for clarity.